

# Ground-based imaging remote sensing of ice clouds: uncertainties caused by sensor, method and atmosphere

Tobias Zinner[1], Petra Hausmann[2], Florian Ewald[3], Luca Bugliaro[3], Claudia Emde[1], and Bernhard Mayer[1]

[1]Meteorologisches Institut, Ludwig-Maximilians-Universität, München, Germany
[2]Karlsruhe Institute of Technology, IMK-IFU, Garmisch-Partenkirchen, Germany
[3]Deutsches Zentrum für Luft- und Raumfahrt, Oberpfaffenhofen, Germany

*Correspondence to:* T. Zinner (tobias.zinner@lmu.de)

**Abstract.** In this study a method is introduced for retrieval of optical thickness and effective particle size of ice clouds over a wide range of optical thickness from ground-based transmitted radiance measurements. Low optical thickness of cirrus clouds and their complex microphysics present a challenge for cloud remote sensing. In transmittance, the relationship between optical depth and radiance is ambiguous. To resolve this ambiguity the retrieval utilizes the spectral slope of radiance between 485 -
5  560 nm in addition to the commonly employed combination of a visible and a shortwave infrared wavelength.

An extensive test of retrieval sensitivity was conducted using synthetic test spectra in which all parameters introducing uncertainty into the retrieval were varied systematically: unknown ice crystal habit and aerosol properties, instrument noise,. calibration uncertainty and the interpolation in the lookup table required by the retrieval process. The most important source of errors identified are uncertainties due to habit assumption: Averaged over all test spectra, systematic biases in the effective
10  radius retrieval of several micrometer can be caused. The statistical uncertainties of any individual retrieval can easily exceed 10 $\mu$m. For optical thickness biases are mostly below 1, statistical uncertainties are in the range of 1 to 2.5.

For demonstration and verification the retrieval is applied to observations by the LMU hyperspectral imager specMACS at the Schneefernerhaus observatory (2650m a.s.l) during the ACRIDICON-Zugspitze campaign in September and October 2012. Results are compared to MODIS and SEVIRI satellite based cirrus retrievals. Considering the identified uncertainties for our
15  ground based approach and for the satellite retrievals, comparison shows good agreement within the range of natural variability in the direct surrounding.



# 1 Introduction

Clouds play an important role in Earth's energy balance as they interact with solar and terrestrial radiation. Cloud feedbacks on climate change and aerosol-cloud interactions remain the largest uncertainties in climate prediction (IPCC, 2013). The particular importance of ice clouds in the climate system has been recognised since a long time (Liou, 1986; Stephens et al., 1990), but their radiative effect is still poorly quantified (Baran, 2012).

Cirrus clouds are usually optically thin clouds in the upper troposphere composed of ice crystals of various complex shapes. Satellite observations reveal that cirrus clouds cover an average area of about 20% of the mid-latitude regions (Sassen et al., 2008) and more than 50% of the tropics (Wylie et al., 1994). The cloud net radiative forcing is determined by the relative contribution of cloud's solar albedo effect and its infrared greenhouse effect, which depend on the cloud scattering and absorption characteristics (Liou, 1986). The net forcing of thin cirrus clouds is assumed to be positive, implying a net gain of energy in the atmosphere (Hartmann et al., 1992).

Cloud properties required to quantify the cloud radiative impact are effective particle size, i.e., the area-weighted mean particle diameter and optical thickness as measure of extinction in the cloud (Thomas and Stamnes, 2002). Several definitions of effective size can be found for non-spherical particles (e.g. McFarquhar et al., 1998). In this study we have chosen the one following Yang et al. (2005).

Cirrus clouds present a challenging task in cloud remote sensing due to their complex microphysics, their high spatial and temporal variability, and their low optical thickness. Compared to optically thick clouds, detection of ice clouds from space is difficult as only a small amount of solar radiation is reflected and the influence on thermal radiation can be weak too. The detectability of optically thin cirrus often relies on information about the surface albedo and surface emission, especially for satellite based methods (Bugliaro et al., 2012). Significant progress in spaceborne cirrus cloud observation has been achieved using active remote sensors (radar and lidar) on CloudSat and CALIPSO satellites, which provide detailed profiles of cloud optical properties, especially for cirrus (Sassen et al., 2008). Apart from the specific active method's limitations (e.g., limited sensitivity of lidar methods to optical thickness larger than 3, limited radar sensitivity to small particles), these methods are limited in spatial coverage and temporal resolution.

The common retrieval technique to derive cloud effective radius and optical thickness from passive space- and airborne multi-channel measurements uses reflected solar radiation at two wavelengths (Nakajima and King, 1990). This approach can be adapted for the retrieval of ice cloud properties (e.g. King et al., 2004). Since reflected and transmitted radiance provide little or no information concerning particle shape, each ice cloud retrieval is based on uncertain assumptions about particle shape (Comstock et al., 2007). These assumptions are known to introduce a major uncertainty in ice cloud property retrievals (e.g. Key et al., 2002; Eichler et al., 2009). Satellite observations have many advantages, such as their global coverage, but they introduce additional uncertainties for cloud retrievals due to limited spatial resolution (hundreds of meters to kilometers). More detailed ground-based cloud remote sensing methods allow to verify and improve space-based techniques. Several ground-based techniques exist for the retrieval of cirrus properties using active (e.g. Wang, 2002; Szyrmer et al., 2012) as well as



passive remote sensing instruments (e.g. Barnard et al., 2008). An intercomparison of ice cloud retrieval algorithms is given in Comstock et al. (2007).

In recent years hyperspectral instruments have become available to atmospheric science and only few approaches exist to exploit these novel possibilities in cloud remote sensing today. An example for such a sensor is the Spectral Modular Airborne

Radiation measurement sysTem (SMART). This non-imaging sensor was used by Eichler et al. (2009) to retrieve properties of cirrus clouds with an optical thickness range of 0.1 - 8 from an airborne persepctive, i.e., from reflected radiances. For ground-based measurements, with the Solar Spectral Flux Radiometer (SSFR; non-imaging; Pilewski et al., 2003), a method to derive optical thickness and effective radius of liquid water clouds with optical thickness ranging from 5 to 100 is presented in McBride et al. (2011). Their lower optical thickness limit is caused by missing observation sensitivity for thin clouds. In

contrast to reflectance-based retrievals, there is no unambiguous mapping between transmittance (transmitted radiance) and optical thickness. Transmittance first increases with cloud optical thickness and then decreases if optical thickness exceeds a critical value. Unfortunately, situations with low optical thickness around 5 are most important for ice clouds. Sassen and Comstock (2001) and Chang and Li (2005) show that most ice clouds have optical thickness in this range. Consequently, this ambiguity has to be solved for observation of cirrus clouds. Schäfer et al. (2013) use an imaging spectrometer in the visible

wavelength region for measurements of optical thickness and solve the ambguity problem by adding additional observer and lidar information. Recently LeBlanc et al. (2015) presented a possible solution for retrieval of optical thickness and effective radius for the pointing SSFR system not providing any imagery. We will present a combination of both, a solution for the transmittance ambiguity and results for image measurements which provide context information on the distribution of optical thickness and effective radius over a large area.

To this end, a new ground-based remote sensing system at the Meteorological Institute Munich is used: specMACS is an imaging spectrometer that provides continuous spectral radiance measurements in the wavelength range 400 - 2500 nm. During the ACRIDICON-Zugspitze campaign in Germany in September/October 2012, a test bed for the later ACRIDICON/CHUVA airborne campaign (Wendisch et al., 2016), measurements of ice clouds above the high altitude environmental research station UFS (German: Umweltforschungsstation) Schneefernerhaus at Mount Zugspitze were collected. In addition to the introduction

of the new retrieval, the general sensitivity of transmittance based ice cloud retrievals is evaluated in detail with respect to the unknown true cloud microphysical situation (particle habit), uncertainties in necessary additional information (aerosol, albedo), instrument accuracy (noise, calibration) and accuracy of the applied retrieval technique itself (interpolation in lookup tables).

The performance of the new retrieval is tested systematically in several sensitivity studies using a large set of synthetic observations. Selected ice cloud observations with specMACS at UFS are analysed with the developed cirrus cloud retrieval

method and compared to results of simultaneous satellite observations from METEOSAT-SEVIRI and MODIS.



## 2   Methods

### 2.1   Hyperspectral imaging spectrometer

The specMACS instrument (Ewald et al., 2015) is part of the Munich Aerosol Cloud Scanner (MACS) instrumentation which is used to investigate cloud-aerosol interactions in the atmosphere. Equipped with high spectral and spatial resolution the instrument is designed to measure solar radiation that is reflected from, or transmitted through, clouds. It consists of two hyperspectral line cameras: the VNIR camera covers the visible and near-infrared wavelength spectrum between 400 nm and 1000 nm, while the SWIR camera measures solar radiation from 1000 nm onwards to 2500 nm. Both systems were manufactured by Specim Ltd., Finland. At a given time the system measures the spectral distribution of solar radiation for a single spatial line. Spectral resolution is 2.5 nm to 4 nm in the visible region and about 7.5 to 12 nm in the near-infrared region (see also Fig. 3). Characterization of all details and instrument calibration can be found in Ewald et al. (2015). For the measurements at the UFS site specMACS was mounted pointing upwards, with the line of sight either oriented perpendicular or parallel to the scattering principal plane, i.e., the solar azimuth angle. The central pixel of the spatial sensor dimension is directed toward zenith, corresponding to a viewing zenith angle of $\theta_v = 0°$. An image is obtained through cloud motion.

### 2.2   Radiative transfer simulation

#### 2.2.1   Radiative transfer model

Radiative transfer was simulated using the radiative transfer code DISORT (Stamnes et al., 1988, 2000; Buras et al., 2011) provided within libRadtran (library of radiative transfer; Mayer and Kylling, 2005; Emde et al., 2015). DISORT (discrete ordinate technique) is applicable for one-dimensional plane-parallel radiative transfer simulations. Within this study, this is chosen as approximation valid for horizontally homogeneous, thin cirrus clouds. We have used the C-code version of DISORT (Buras et al., 2011) which is included in libRadtran. This version includes an intensity correction method which is especially useful for the simulation of highly peaked phase functions which are typical for ice clouds. The method can handle phase functions stored on an arbitrary scattering angle grid (for our simulations this grid contained 498 angles). The number of streams for DISORT calculations was set to 16.

Absorption by atmospheric gas molecules is parametrised using the representative wavelength parameterisation REPTRAN (Gasteiger et al., 2014) which is part of libRadtran. Spectra were calculated at high wavelength resolution and convolved with the sensor's spectral sensitivity (Ewald et al., 2015). Ground-based measurements of transmission through ice clouds are simulated in terms of spectral transmittance $T$, defined as

$$T_\lambda = \frac{\pi L_\lambda}{E_{0,\lambda}\cos\theta_0}, \tag{1}$$

where $L_\lambda$ is transmitted spectral radiance, $\theta_0$ solar zenith angle and $E_{0,\lambda}\cos\theta_0$ is the incident extraterrestrial irradiance at top of atmosphere.



### 2.2.2 Optical properties of ice clouds

To perform realistic radiative transfer calculations in a cloudy atmosphere, models of cloud bulk microphysical and optical properties are essential.

For the simulations we have used the HEY parameterisation which is available in libRadtran (Emde et al., 2015). This parameterisation is based on single scattering properties of ice crystals (Yang et al., 2013). To generate bulk optical properties the parameterisation assumes gamma size distributions with parameters typical for ice clouds. The parameterisation is available for six individual habits and for a general habit mixture modeled after the one defined by Baum et al. (2005). To quantify the retrieval sensitivity to habit mixture assumption (Sec. 3.5), radiative transfer is simulated for ice clouds consisting of particles of an individual of the six different habits and of the 'Baum-like' mixture. Phase function examples for six habits and the resulting habit mixture are illustrated in Figure 1.

### 2.2.3 Surface albedo

Due to multiple scattering between ground and cirrus clouds, a spectrally resolved surface albedo is necessary to simulate hyperspectral transmittance. The discrete channels of the MODIS surface albedo product (Strahler et al., 1999) are fitted using a linear combination of albedo data from the ASTER spectral library (Baldridge et al., 2009). For the measurement site at UFS, the smallest residuals in this least-square-fit are obtained by combining the spectral albedo of grass and limestone as shown in Fig. 2. MODIS white-sky albedo acquired between 5 - 21 September 2012 is averaged over an area of $20{\times}20$ km$^2$ surrounding the site. This area corresponds to more than 50% of the radiance reaching cloud bottom from below at 9 km height directly above the ground site after Lambertian reflection at the ground.

## 3 Retrieval of optical thickness and effective radius

### 3.1 Idea

According to the fundamental work of Nakajima and King (1990), water cloud properties can be derived from satellite-based measurement of cloud reflectance at two wavelength bands. We apply this approach to ground-based measurements of ice cloud transmittance. Remote sensing of radiative properties of ice clouds is complicated due to their low optical thickness (mostly smaller than 5), non-spherical particle shape, and possible particle orientation. In the visible spectral range, transmittance spectra are dominated by scattering of clouds, aerosols, and gas molecules and by surface reflection. At longer wavelengths, in the near-infrared, liquid water absorption increases strongly. Consequently, transmittance is especially sensitive to cloud optical thickness in the visible spectrum (Fig. 3a) and to effective particle size in the near-infrared spectral range (Fig. 3b). This correlation is exploited to retrieve ice cloud properties from spectral measurements. Analogous to the approach of Nakajima and King (1990), radiances at 550nm and 1600nm are stored in a lookup-table as function of optical thickness and effective particle radius (Fig. 4) which can be compared to measurements. In contrast to reflectance, a wide range of this diagram does not show an unambiguous relationship with optical thickness as transmittance first increases and then decreases with increasing





optical thickness. The presence of an optically thin cloud increases transmittance by cloud particle scattering compared to the dark diffuse clear sky (Fig. 3a, solid lines). Successive increase of ice cloud optical thickness leads to increased scattering, but also to a growing degree of cloud scattering and absorption. If optical thickness exceeds a critical value, cloud reflection back to space and absorption become dominant and transmittance decreases (Fig. 3a, dashed lines).

A method has to be found to resolve the ambiguity in transmittance lookup tables and to separate overlapping regimes of optical thickness in Figure 4. An important factor in distinguishing cloud optical thickness is sky colour. As optically thin clouds are partially transparent, the short-wave "blue" contribution from atmospheric Rayleigh scattering is visible. With increasing optical thickness the cloud scattering becomes dominant, leading to a "grey" spectrally invariant appearance. A measure for this colour is the slope of transmittance spectra $T_\lambda$ in the visible spectral range (VIS), defined as

$$S_{\mathrm{VIS}} = \frac{100}{T_{550}} \cdot \left. \frac{dT_\lambda}{d\lambda} \right|_{\lambda_1}^{\lambda_2}. \tag{2}$$

The derivative of transmittance is computed by linear regression in the range from $\lambda_1 = 485$ nm to $\lambda_2 = 560$ nm normalised by transmittance at 550 nm and scaled with 100 to obtain values in a range comparable to transmittance. Small optical thickness is associated to negative $S_{\mathrm{VIS}}$ (high fraction of blue), while $S_{\mathrm{VIS}}$ becomes neutral or positive (low fraction of blue) for larger optical thickness. The wavelength range to calculate $S_{\mathrm{VIS}}$ is chosen because (1) it is part of the visible spectrum which is

sensitive to optical thickness, (2) it exhibits a smooth spectral trend which allows for determination of a slope (cf. Fig. 3a) , and (3) it is not too close to the lower end of the spectral range of specMACS where sensor sensitivity and calibration accuracy quickly deteriorate. $S_{\mathrm{VIS}}$ allows to separate overlapping parts in Figure 4 resulting in two widely unambiguous parts of the lookup table.

In order to exploit the information of the visible spectral slope $S_{\mathrm{VIS}}$, the lookup table is expanded with the slope as a

third dimension. This three-dimensional approach is illustrated in Figure 5. It resolves the ambiguity in conventional two-dimensional lookup tables for transmittance (Fig. 4). Transmittance at 550 nm and 1600 nm represent the two "classical" dimensions of the lookup table. Additionally, spectral slope $S_{\mathrm{VIS}}$ is used as a third. In this 3D diagram the simulated radiation values span a non-intersecting surface: points associated to larger optical thickness values are shifted to larger spectral slope values along the z-axis. Consequently, it is now possible to unambiguously match a pair of optical thickness and effective

radius to a measured transmittance spectrum in the 3D lookup table.

## 3.2 Lookup table generation

The basis of the introduced retrieval method is a large set of simulated ice cloud transmittance for all needed specMACS bands which are arranged in lookup tables. In this section we describe the setup for these simulations. A standard summer mid-latitude atmosphere by Anderson et al. (1986) with an ozone column scaled to 300 DU is used. The extraterrestrial solar

spectrum is taken from Kurucz (1992). A horizontally homogeneous ice cloud layer is located at a base height of 9 km with a geometrical thickness of 1 km. Ice particles are assumed to be randomly orientated. The properties of this model ice cloud are varied as follows:



- – 45 effective radius values $r_{\text{eff}} \in$ [5 - 90 µm]; increments of $\Delta r_{\text{eff}}$=1 µm between 5 and 30 µm, $\Delta r_{\text{eff}}$=2.5 µm between 30 and 65 µm, $\Delta r_{\text{eff}}$=5 µm between 65 and 90 µm

- – 60 optical thickness values $\tau \in$ [0 - 20]; increments of $\Delta\tau = 0.1$ between 0 and 1.5, $\Delta\tau = 0.25$ between 1.5 and 10, $\Delta\tau = 1$ between 10 and 20

- – 6 habits (solid column, hollow column, six-branch rosette, plate, droxtal, roughened aggregate) and the described habit mixture

Solar zenith angles $\theta_0$ are simulated for the range of 24° to 75° in increments of 1°. Viewing zenith angles $\theta_v$ from 0° - 10° in increments of 1° are computed according to the sensor's field of view during the measurements. As radiative transfer is simulated for one-dimensional horizontal homogeneous clouds, specification of relative azimuth angle $\phi_{rel} = \phi_v - \phi_0$ is sufficient. Three relative azimuth angles are taken into account: $\phi_{rel} = 0°$ and $\phi_{rel} = 180°$ (spatial sensor line parallel to principal plane) or $\phi_{rel} = 90°$ (perpendicular to principal plane).

Aerosol plays a minor role for the UFS observations evaluated here due to the elevated height above the boundary layer. Nevertheless, the effect of a moderate aerosol layer on the retrieved values was studied. Two settings are considered: one without aerosol and one assuming OPAC continental-average aerosol as provided in libRadtran (Emde et al., 2015). This aerosol mixture is based on microphysical properties of various aerosol types included in the OPAC database (Hess et al., 1998). The aerosol optical thickness was set to 0.2 at 550 nm which is a typical value for the Meteorological Institute site in Munich. See Fig. 3 for results on specMACS spectral resolution.

### 3.3 Cloud phase detection

As a first step in any ice cloud retrieval, the observed cloud thermodynamic phase has to be determined. Ice crystals and liquid droplets exhibit different absorption coefficients and consequently different cloud radiative impact. Particle absorption is described by the imaginary part of the complex refractive index which depends on wavelength and phase. For cloud side remote sensing, Zinner et al. (2008) and Martins et al. (2011) proposed to use the ratio of reflectance in narrow spectral regions at 2.1 and 2.25 µm to separate between ice and liquid water. A similar approach is applied here for measurements of cloud transmittance. Ice cloud transmittance rises strongly in the wavelength range from 2.1 to 2.25 µm as absorption decreases in this spectral region. In contrast, liquid water transmittance changes only slightly in the same range corresponding to a nearly constant absorption coefficient. A near-infrared (NIR) ratio $I_{\text{NIR}}$ can be defined as $I_{\text{NIR}} = T_{2.1\mu\text{m}}/T_{2.25\mu\text{m}}$, with transmittance $T_{2.1\mu\text{m}}$ at 2.1 $\mu m$ and $T_{2.25\mu\text{m}}$ at 2.25 $\mu m$. As the NIR-ratio is generally smaller for ice clouds than for water clouds, cloud phase may be determined by means of a threshold in $I_{\text{NIR}}$. We assume that for $I_{\text{NIR}} < 0.92$ the observed cloud consists of ice particles and for $I_{\text{NIR}} \geq 0.92$ the observed cloud might contain liquid water droplets. Tests using simulated data show a good performance of this NIR-ratio method (see 3.5), but the adequate threshold in $I_{\text{NIR}}$ slightly depends on assumed ice particle habit and viewing geometry.





### 3.4 Retrieval procedure

The algorithm to retrieve cloud optical thickness and effective radius from a measurement is performed in five steps:

1. Detection of cloud phase using the NIR-ratio.

2. Interpolation of an applicable lookup table from the nearest tabulated sun geometries' values.

3. Determination of distance of measurement to all lookup table elements.

4. Selection of subset of close elements in lookup table.

5. Calculation of weighted average over this subset.

After detection of ice phase an applicable lookup table is obtained for the sun and sensor geometry ($\theta_v$, $\theta_0$ and $\phi_{\mathrm{rel}}$) given for the measurement situation. To this end, lookup table elements are interpolated from the nearest simulated observation geometries' values. For four nearest solar zenith and four nearest viewing zenith angles, the three retrieval parameters (transmittance at 550, 1600 and slope 485-560 nm) are interpolated using with respect to scattering angle.

In the third retrieval step, the three-dimensional distance $\delta T_{LUT,i}$ between one measurement and all points $i$ in the interpolated lookup table are calculated as

$$\delta T_{LUT,i} = \Big[(T_{550} - T_{550,i})^2 + (T_{1600} - T_{1600,i})^2 + \tag{3}$$
$$(S_{\mathrm{VIS}} - S_{\mathrm{VIS},i})^2\Big]^{1/2}$$

with transmittance $T_{550}$ at 550 nm, transmittance $T_{1600}$ at 1600 nm, and visible spectral slope $S_{\mathrm{VIS}}$, each for the measured data as well as for points $i = 1, ..., N$ in the lookup table. The distance $\delta T_{LUT,i}$ is directly obtained from the given values of transmittance and transmittance slope, because the overall range of values for all three measured parameters is comparable (compare Fig. 5).

A maximum distance accepted is set to $\delta T_{\max} = 0.1$. That means, if no points in the lookup table are found within $\delta T_{\max}$, the retrieval fails. Cases for which more than three values are found, the threshold is lowered stepwise to $\delta T_{\max} = 0.05, = 0.025, = 0.0125$. Especially for low optical thickness, where many possible solutions lie within a small transmittance range, this improves retrieval accuracy. The distance to the closest point in the lookup table $\delta T$ is stored as a measure of retrieval quality. Quality $1 - \delta T / \delta T_{\max} = 0$ is related to the maxium search radius, while larger values are related to better matches and a perfect match would be quality 1.

Finally, the ice cloud properties $x$, optical thickness and effective radius, are retrieved by taking a distance weighted average over the remaining elements in the chosen lookup table subset:

$$x_{\mathrm{ret}} = \begin{cases} \dfrac{\sum\limits_{i=0}^{N} w_i\, x_i}{\sum\limits_{i=0}^{N} w_i} & \text{if } \delta T_{LUT,i} < \delta T_{\max} \\[2em] x_i & \text{if } \delta T_{LUT,i} = 0 \end{cases} \tag{4}$$



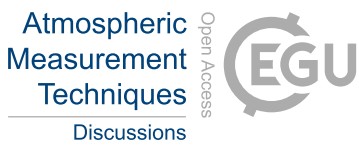

with

$$w_i = 1/\delta T^4_{LUT,i} \tag{5}$$

The weighting factor $w$ is found along the following line of thought. Considering a 3D space filled with uniformly distributed points, the number of points contributing to an averaged result "increases proportional to the surface area $\mathrm{d}A = 4\pi\delta T^2$ with

increasing distance $\delta T$. To account for this factor of $\delta T^2$, the weight for each point has to be at least $1/\delta T^2$. For more emphasis on close points a smaller weight should be chosen. We use $w = 1/\delta T^4$.

### 3.5 Sensitivity tests

A large set of simulated cirrus transmittance spectra is used in seven retrieval test cases in order to determine typical uncertainties inherent to the presented retrieval method (and to other similar ice cloud retrievals based on transmittance or reflectance).

The cloud properties retrieved from these synthetic cases may directly be compared to the corresponding simulated parameters similar to the approach in Bugliaro et al. (2011). Individual influences on retrieval accuracy can be isolated this way: Test case 1 is a consistency check whether simulated transmittance from the lookup table is related to the correct tabulated retrieved properties by the retrieval procedure. Using radiative transfer simulations not included in the lookup, test cases 2 and 3 examine retrieval sensitivity with respect to the necessary interpolation of values of effective radius and of optical thickness (case 2) and

of observation angle (case 3) in the solution space. The assumption of a fixed crystal mixture is tested against several single habit situations (case 4). Aerosol concentration is varied between "no aerosol" (representative for the high altitude measurements at UFS) and "typical aerosol" (for the area around Munich, case 5). Impact of instrument noise (case 6) and calibration accuracy (case 7) is also evaluated applying typical values (Ewald et al., 2015) . The results are summarised in Table 1.

Three error measures are provided: bias $B_x = \langle x_{\mathrm{ret}} - x_{\mathrm{true}} \rangle$, root mean square error (RMSE), $R_x = \sqrt{\langle (x_{\mathrm{ret}} - x_{\mathrm{true}})^2 \rangle}$,

and the 95 percentile $P_{x,95}$ of all absolute errors. $x_{\mathrm{ret}}$ and $x_{\mathrm{true}}$ are retrieved and "true" quantity, $x$ being either $\tau$ or $r_{\mathrm{eff}}$), $\langle ... \rangle$ is the average over all retrieval instances. In addition, an error rate $F$ is defined. It gives the percentage of "incorrect" retrieval cases in the total number of performed retrieval tests. A test case is perceived as incorrect if the deviation of retrieved and true quantity is larger than the critical values $|\Delta r_{\mathrm{eff}}| > 5$ µm and $|\Delta\tau| > 1$ for any one of the two quantities.

More than 30,000 tests are run for each of the following cases. Random combinations of spectra for 26 values of $\tau$ in

the range of 0.3 - 17, 17 values of $r_{\mathrm{eff}}$ within 8 - 70 µm, seven different solar zenith angles $25° < \theta_0 < 58°$, two relative azimuth angles $\phi_{rel} = (0°, 180°)$, and five viewing zenith angles $1° < \theta_0 < 7°$ are used. In a first case, the retrieval is applied to spectra which are part of the lookup table for the assumed habit mixture. As anticipated, simulated cloud properties are exactly reproduced and all error measures assume a value of zero.

For test 2 the retrieval is applied to a large set of additional synthetic observations simulated for values of effective radius

and cloud optical thickness that are not part of the lookup tables: In order to limit the number of necessary tests while not damaging the vailidty of results, values are chosen in the following way: Within the range covered by the variable lookup table sampling grid $\Delta x$ ($x = \tau$ and $x = r_{\mathrm{eff}}$, values see section 3.2), random values are chosen at a distance of $0.25 \times \Delta x$ from tabulated lookup table grid points. This is the average distance from grid points for random measurements. Each of these ice





cloud test cases is simulated for 20 geometries included in the lookup tables assuming the given habit mixture. This test isolates the effect of interpolation of measurements within the given lookup table grid: The root mean square errors are comparable to the resolution of the lookup table (in case of optical thickness) or even better (for effective radius). It is apparent that the error frequency distributions are not Gaussian, but narrow distributions with a few outliers. Thus, the 95 percentile $P_{x,95}$ for

both quantities is similar to $R_x$. That means, $R_x$ is strongly influenced by the outliers, while the vast majority of deviations is smaller. Furthermore 5.7% of retrieval errors are larger than the critical values $|\Delta r_{\mathrm{eff}}| > 5$ µm or $|\Delta \tau| > 1$.

In test 3, solar and viewing zenith angles used for the test observations are not part of the lookup tables either. Thus interpolation of tabulated values has to be used on illumination geometries as well. Now test cases for the involved angles are chosen at locations with a distance of $0.25 \times \Delta x$ (with $x$ being solar or viewing zenith angles). Azimuth angle is not varied, as we

assume that orientation using the sun is precise for quick measurements. Cloud parameters are varied as described for test 2. $R_r$ and $P_{r,95}$ are only slightly larger than in test 2 for $r_{\mathrm{eff}}$, hardly any changes are detected for $\tau$. Error rate $F$ increases to 8%.

One major uncertainty in all ice cloud property retrievals is introduced by assumption of a specific ice crystal habit mixture. In test 4, the retrieval based on habit mixture is applied to synthetic observations simulated using single habit characteristics. We apply the retrieval to the same test cases as in test 3, using lookup tables for each of the six habits (see Tab. 1). Compared to test

3, bias, uncertainty, and error rate do strongly increase due to the habit mismatch. The optical thickness retrieval is characterised by biases in the range of $\pm 1$. As an example for the effective radius retrieval for one of 70 tested geometries, results for bullet rosettes are shown in Figure 6 with coloured points indicating each of the 442 single tests and lines representing mean, standard deviation, and the interval with 95% of all individual errors. While small effective radius values tend to be overestimated, large values are underestimated for this geometry on average. Effective radius is generally overestimated for small optical thickness

below about 1. For medium optical thickness between 1 and 7, the effective radius is mostly under-estimated while retrievals for optical thickness larger than that have only a small bias. This behaviour can not be generalised, but strongly varies with geometry and ice particle habit. The error rates presented in Table 1 ranging from 48% to 84% are probably an upper limit for the real errors, as clouds rarely consist of a single habit but of a habit mixture probably mitigating the errors shown.

Beside habit assumption and interpolation techniques, aerosol optical thickness influences retrieval performance. In test 5

a retrieval assuming no aerosol is applied to the test cases of test 3, but a continental average aerosol (Hess et al., 1998) with an optical thickness of 0.2 at 550nm was used to simulate the observations The actual increase of 0.5 is larger than the AOD mismatch possibly caused by the fact that an even larger cloud optical thickness is needed to match larger sky brightness by aerosol due to greater forward scattering by large ice particles compared to smaller aerosol size. Not surprisingly, larger values of RMSE and 95 percentile occur for small values of $\tau$, when the aerosol influence on transmittance becomes predominant.

Another factor that influences the retrieval uncertainty is instrument noise (cf. Sec. 2.1). In test 6, random noise is added to the synthetic observations of test 3 with a maximum magnitude of $\pm 0.25\%$ of the signal. This corresponds to a signal-to-noise-ratio (SNR) of 400:1, which is a typical value found during characterisation of specMACS (Ewald et al., 2015). As expected for random noise, the retrieval bias is identical to test 3, but RMSE and false detection rate are slightly increased.

More important than the effect of instrument noise is the calibration accuracy that can be expected. A 5% offset due to

calibration is assumed for tests 7a and 7b, which is a conservative estimate for specMACS for most of the covered spectrum




(Ewald et al., 2015). Biases introduced are small while uncertainties, especially for effective radius are much larger. Overall the effect is similar to the variation in aerosol content. No bias is found for the effective radius retrieval. Uncertaintiy values RMSE at 6 μm and $P_{r,95}$ at 12 μm show that this has to be the effect of a cancellation of opposing contributions.

Further uncertainty is introduced by solar zenith angle determination, alignment of the sensor, and changes of the sun position during the measurement. However, we expect those to be much smaller than the uncertainties discussed above. Besides, transmittance spectra are influenced by water vapour concentration in the atmosphere. For the wavelengths used in the retrieval, which are not located close to any strong water vapour absorption bands, only small differences in transmittance are to be expected. Water vapour column variations between 5 - 50 kg m$^{-2}$ (a range as large as the difference between polar and tropical regions) only cause differences in the order of 0.1% in transmittance.

The detection of ice phase worked for the vast majority of all considered test cases. Between 10 and 700 test cases out of 30,000 tests were incorrectly classified as liquid water throughout the 13 tests listed in Table 1; on average about 170 or 0.6% of each case. These misclassifications are mostly related to small optical thickness around 0.5. For the ice habit tests 4a to 4f as well as 7a and 7b, up to a few 100 of 30,000 test retrievals did not succeed because no lookup table values were found near the measurement using $\delta T_{\max} < 0.1$.

## 4   Application and comparison to satellite data

In this section retrievals from the presented ground-based method are presented for two cases during the ACRIDICON-Zugspitze measurement campaign at the research station UFS Schneefernerhaus in October 2012. The ice cloud retrieval was applied to three specMACS data sets collected on 2 October 2012 at 8:08-8:21 UTC, 8:57-9:10 UTC and 9:53-10:04 UTC and two data sets collected on 3 October 2012 at 14:47-15:08 UTC and 15:09-15:14 UTC. Retrieval results are compared to satellite products from METEOSAT-SEVIRI and MODIS. On both days SEVIRI Rapid Scan data over the Zugspitze area were evaluated using the DLR APICS retrieval (Algorithm for the Physical Investigation of Clouds with SEVIRI; Bugliaro et al., 2011). On 2 October a TERRA overpass at 10:20 UTC almost perfectly matched the time of the third specMACS measurement interval and comparison to MODIS collection 6 data (Platnick et al., 2015a) is possible. While APICS is used with a habit mixture following Baum et al. (2005), the MODIS Collection 6 retrieval assumes severely-roughened aggregated columns (Platnick et al., 2015b).

Figure 7 shows the weather situation as seen by SEVIRI. Stationary orographic low level cloudiness, recognisable as yellow area in the false-colour-composite, is visible at the Alps (south of UFS position, red cross). On 2 October it is concentrated at the northern edge of the mountains close to the Zugspitze and east of it; on 3 October low level cloudiness is shifted southwards. Bands of blueish synoptic scale cirrus cloudiness not related to the lower clouds move through the area from West to East on both days. On 2 October, the filaments of the large North-South oriented band that just passed Zugspitze were observed by specMACS a few minutes up to two hours before the MODIS image collection time; on 3 October measurements were collected with a more homogeneous cirrus deck above Zugspitze.



### 4.1   3 October

The later less complex case is discussed first. For two specMACS data sets collected between 14:47 and 15:13 UTC on 3 October 2012 a comparison to METEOSAT-SEVIRI data is possible.

Figure 9 shows a part of the specMACS data collected at 3 October 2012 14:47-14:57 UTC. The specMACS measurement was set up pointing vertically with the sensor's spatial line of measurements perpendicular to the solar azimuth angle of 243° at time of measurement (sun at South-West after local noon). Data collected by specMACS with angular resolution of about 0.05° (18° field-of-view, 320 spatial pixels) and time resolution of 4 frames-per-second is regridded to approximately fill a 30 m × 30 m grid using the wind speed of 18.5 m/s (Fig. 12a) for a cloud bottom at 6500 m above UFS (cf. Fig. 8). For the case shown, the given wind direction is almost aligned with the sun at South-West. That means, advection was nearly perpendicular to the sensor spatial line and an almost undistorted image of the cirrus is observed by specMACS (Fig. 9). Figure 9b and c show retrievals of optical thickness and effective radius, Figure 9d shows retrieval quality. Retrievals have been possible anywhere in this scene. At quality values above 0.5. It can be seen in Figures 9b and c that optical thickness and effective radius are not totally independent. Though there could be physical reasons for this observation, it is also an inevitable effect of unknown ice crystal habit which leads to slightly varying interdependence of the used spectral channels and retrieved parameters. This is inherent to all passive ice remote sensing methods based on "Nakajima-King style" retrievals.

Figure 10 shows the comparison of data from specMACS and SEVIRI for the UFS position over time. The parallax corrected APICS SEVIRI satellite retrievals available every 5 minutes are marked with green points. The error bars on SEVIRI data show the standard deviation of values in the neighbourhood. This includes all ice retrievals in the 8-connected pixels around the UFS pixel (corresponding to 200 $km^2$). From each 2D scene of specMACS data, the collected data averaged across the spatial dimension and over 4 second segments are shown with red symbols. The standard deviation of all available retrievals in such a 4 second segment leads to the uncertainty estimate labelled with thin red lines. These values correspond to an area of only 0.16 $km^2$. A specMACS scene like Fig.9 contains data from about 10-50 $km^2$ area. Obviously, specMACS observes more extreme values due to its high spatial and temporal resolution while these extremes are averaged out over a SEVIRI pixel of several kilometer size. Comparison of SEVIRI data points to specMACS data at the very same moment shows very small differences. Averaged over all data, optical thickness seems to be slightly lower, effective radius slightly higher for SEVIRI. Agreement is very good to about 0.5 in $\tau$ and a few micrometer in $r_{\text{eff}}$.

### 4.2   2 October

On one hand, the second case allows an additional comparison to MODIS data, on the other hand, it is complicated by low level cloudiness which affect the satellite products. Figure 11 shows MODIS collection 6 products for 2 October at the time of the SEVIRI measurement (10:20 UTC) as presented in Figure 7a: effective radius, optical thickness and cloud phase. At the southern edge of the displayed domain the low level orographic mostly liquid water clouds can be found. Overlaid is the cirrus band moving through the area. Differences between the two cloud types are obvious in optical thickness and effective radius. Water clouds are optically very thick ($\tau > 30$) while optical thickness of ice clouds ranges from 1 to 10. Effective radius





retrievals for water droplets show small values (10-20 μm), ice particle retrievals show values between 25-40 μm and even more than 50 μm around cloud edges. Interestingly the above mentioned interdependence of $\tau$ and $r_{\text{reff}}$ is also visible in this data.

The cirrus band is advected from West to East over the low water clouds with almost no change in shape and appearance (compare METEOSAT and cloud radar data in Figures 7 and 8). Thus we analyzed MODIS data over the full 2 hours time

period assuming a constant advection of the cloud band with wind speed. This approach is illustrated by the red lines of symbols in Fig. 11. The first symbol in the lower left corner marks the position of the specMACS sensor at UFS at the time the cloud band has just passed UFS moving east. Each symbol to West-North-West of this position is related to a time step of 133 seconds or about 1 km distance backward in time along wind direction. This is based on wind direction and wind speed of 8 m/s taken from nearby radiosonde data at cloud height of the cirrus band from cloud radar data at the site: 4 km above UFS

between 8 and 10 UTC (Fig. 8). To the left and right of the wind direction little cross symbols mark an uncertainty region as large as $\pm$ 1 km (MODIS pixel size) at measurement time and growing towards $\pm$ 8 km for a time offset of more than 2 hours (related to a wind direction uncertainty of $\pm 7°$). Water cloud retrievals dominate a large part of the cross section, most likely a consequence of low water clouds forming due to topography below an optically much thinner ice cloud.

Figure 12 shows specMACS data collected at 2 October 2012 8:56-9:10 UTC. In this case, the specMACS measurement

was again set up pointing vertically with the sensor's spatial line perpendicular to the principal plane (sun in the South-East before local noon, $\phi = 142°$). As before, specMACS data is regridded to onto a 30 m $\times$ 30 m grid, this time with wind speed of 8 m/s. The given wind direction South-West means that advection was nearly parallel to the sensor spatial line. Thus the 2D display of the observed cirrus in Figure 12 appears distorted. If this parameter is smaller than zero (grey areas), no tabulated values for transmittance and transmittance slopes were found in the lookup table within the maximum search radius and no

retrieval is performed. Retrievals have been possible almost anywhere in the scene. Mostly at quality values above 0.5. Grey areas label pixels for which no tabulated values of transmittance were found within the maximum search radius and no retrieval is performed.

All three data sets are compared in Figure 13. The MODIS data (black) are taken from the cross section through the 10:20 UTC data set actually observed by MODIS. At the time of the MODIS overflight (10:20 UTC) no ice cloud was found at the

position of UFS neither in MODIS nor in METEOSAT-SEVIRI retrievals. In particular, APICS detected no ice clouds after 10:05 UTC. The error bars on MODIS (black) show the standard deviation of values for all ice retrievals in the surrounding area defined by wind direction uncertainty (Fig. 11a-c).

For optical thickness (Fig. 13a) both satellite based retrievals agree roughly, as far as the overall value range is concerned. Both show a decline in optical thickness from large values around 10-13 towards small values 2-4, although they do not agree

in timing of this decline. specMACS optical thickness agrees well with satellite retrievals around 10 UTC. Not surprisingly the high resolution transmittance retrieval picks up the very small values at the rear end of the cirrus band moving eastward (after 10:00) better than low resolution reflectance retrievals from satellite. MODIS misses them completely (likely due to thick water clouds below), SEVIRI shows larger values around 2. Given the resolution differences, specMACS retrievals are mostly within the expected uncertainty of the satellite data for 8:56-9:10 UTC (both MODIS and SEVIRI) and for 8:08-8:21 UTC

(only MODIS).



For such a cirrus cloud band moving unrelated to lower level cloudiness, optical thickness values around 10 and above - as retrieved by both satellite retrievals - are a sign of the underlying cloud layer's influence. Chang and Li (2005) analyse these cases of thin cirrus overlaid over thicker low level clouds in detail for MODIS data and they find similar differences as seen for this scene. Water cloud and ice cloud optical thickness can not be separated from satellite perspective, while the water clouds

below our observation height (UFS) obviously do not disturb the zenith observation (compare 11d).

Effective radius values are also in good agreement for the time period 9:53-10:04 UTC. The scatter of specMACS retrievals becomes high when optical thickness becomes very low after 10 UTC. While the two satellite retrievals agree remarkably well over the whole time period, specMACS derived particle size is larger for the earliest measurement period 8:08-8:21 UTC. Still the observed deviation seems to be within the range of values explainable by uncertainty introduced through underlying water

clouds with lower effective radius (compare Fig. 13b).

## 5    Summary and discussion

We presented a new method for the retrieval of ice cloud optical thickness and effective radius from spectral measurements of transmitted radiance. Using data from a new spectral imager (specMACS; Ewald et al., 2015) a phase detection and retrieval of optical thickness and effective radius was set up. Phase detection uses a near-infrared ratio of transmitted radiances ($T_{2.1\mu m}$

/ $T_{2.25\mu m}$). It is combined with a well known two wavelength approach following, e.g., Nakajima and King (1990). An important extension of the method is the addition of a third parameter to the retrieval to resolve the ambiguity of transmittance retrievals with respect to increasing optical thickness. If optical thickness increases, first diffusely transmitted downward radiance increases. For increase above optical thickness of 4 to 6 radiance values decrease again. Interpretation of the spectral slope observed at 485 - 560 nm overcomes this problem as it steadily decreases from positive values (blue sky color) to neutral

values (grey sky color) for increasing optical thickness in this region. This slope is combined with radiances at 550 nm (mainly sensitive to scattering and optical thickness) and $1.6\mu m$ to create a retrieval using a lookup table based on 1D radiative transfer simulations.

A second important part of the presented work is the rigorous test of sensitivities of the established retrieval to the method's internal accuracy as well as its sensitivity to unknowns and uncertainties of real world observations. In general, for this type of

ice cloud remote sensing instrument accuracy (absolute calibration and noise), interpolation in the given lookup table's forward solutions, additional scattering by unknown aerosol and unknown ice crystal habit mixture in the observed cloud have to be considered as sources of bias and uncertainty. Some of these are methodical issues. The issue of unknown habit is of course a core part of the physics of our objects of study. Still it is an unsolved issue in the remote sensing community. The pre-calculated forward solutions of the retrieval have to assume a certain ice particle habit or mixture situation which most likely is not the

one present for a given observation.

Using more than 300,000 synthetic measurements over the expected range of optical thickness, effective radius and observation geometries, it was possible to isolate effects of these uncertainties systematically. They show that uncertainties due to lookup table interpolation stay minimal: systematic deviations of optical thickness from true values are negligible and below



0.5 μm for effective radius. Still the variation of a specific retrieval can be in the range of 0.5 for $\tau$ and a few micrometer for effective radius. Instrument noise creates a comparably small effect. Also unknown aerosol for moderate aerosol optical conditions cause systematic biases around 0.5 in $\tau$ and 0.5 μin $r_{\text{eff}}$. Uncertainties of effective radius can grow to about 10 $\mu m$. The impact of limited calibration accuracy is likely to be in a similar range. Errors can be positive or negative by more than 5 $\mu m$ for $r_{\text{eff}}$ and around 1 for $\tau$, strongly depending on the optical thickness regime. The biggest problem for retrieval accuracy is obviously unknown habit. Especially for effective radius it causes systematic errors much larger than all other influencing factors. Biases are in the order of 5 $\mu m$ for $r_{\text{eff}}$ and for $\tau$ in the order of 1. Uncertainty of a specific retrieval can easily be wrong by more than 10 $\mu m$ and 1 to 2 for $\tau$.

Our application to two cirrus cases observed during the ACRIDICON-Zugspitze campaign in fall 2012 agreement within the expected uncertainty ranges. Given that (1) crystal habit assumptions are not the same for the three compared methods, (2) reflectance based methods suffer from additional uncertainty due to strong surface and cloud influence from below the cirrus clouds, and (3) resolution of the measurements is very different, the deviation of average optical thickness values and effective radius is mostly within the observed variability in the surrounding area.

Obviously there are limitations to passive remote sensing of thin ice clouds. Chang and Li (2005) show that thin cirrus ice clouds have a long term and large scale average optical thickness in the range of 1 to 2 with standard deviation in the same range. In this respect the presented results seem to showcase a rather typical situation. Given these average values, the possible optical thickness errors of mean values from the synthetic tests up to values of 1 and the observed differences of mean values up to 4 (though mostly much less) demonstrate the full challenge of ice cloud remote sensing.

The uncertainty values found here for the ground-based perspective are in good agreement with existing results on the satellite perspective. Key et al. (2002) anticipated differences of more than 50% in optical thickness retrievals for various habit assumptions based on simple radiative transfer considerations in the literature. In a case study for an airborne reflectance based retrieval Eichler et al. (2009) found relative differences of around 50% in optical thickness and up to 20% in effective radius depending on habit assumptions.

In order to improve the situation some of the mentioned uncertainties should be reduced. Ground-based high spatial resolution remote sensing of ice clouds in itself already reduces some uncertainties of satellite retrievals. Ground-based methods are much less affected by surface albedo. Low level clouds are easily distinguished from cirrus and, in case of elevated observations, can be excluded almost entirely. Due to high spatial resolution separation of clear and cloudy areas is more precise while sub-pixel inhomogeneity and clear sky "contamination" affects satellite data. The higher resolution and reduced albedo sensitivity improve the sensitivity to small optical thickness values. It has to be admitted though that 3D effects, i.e. effects of horizontal radiation transport, are more likely to affect retrievals at higher spatial resolution.

One important technical approach is the reduction of calibration inaccuracies. For example, LeBlanc et al. (2015) demonstrate retrieval approaches based on normalized radiance instead of absolute measurements. For their solar spectral flux radiometer (SSFR), operating at a similar wavelength range as specMACS (though without imaging capability), they normalize all spectral radiance values by the value of one spectral channel. Dependence on calibration accuracy is much reduced this way, because only the more stable channel-to-channel accuracy has to be considered instead of absolute radiometric accuracy



which is much harder to achieve. Of course the most important step forward would consist in a reduction of the crystal habit uncertainty. To this end, efforts could include the use of the presence of optical scattering phenomena like type and intensity of halo displays when possibly. Different phenomena are related to specific particle shapes and orientation and their intensity includes information on the mixture with less perfect rough ice particles. A combination of the presented method with additional

5  information of this kind will be the next step in our effort to provide better ice cloud property observations.

*Acknowledgements.* We thank Hong Gang for providing the single ice crystal properties from Yang et al. (2013). This work was partly funded by DFG, German Research Foundation projects MA 2548/6-1, MA 2548/9-1, INST 86/1256-1.



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



**Table 1.** Results of sensitivity studies: retrieval bias ($B_\mathrm{r}$, $B_\tau$), root mean square error ($R_\mathrm{r}$, $R_\tau$), 95 error percentile ($P_{\mathrm{r},95}$, $P_{\tau,95}$), false detection rate $F$, habit of simulated test cases, and details about chosen test cases. Optical thickness of tested cases 0.3-17 and effective radius 8-70 [µm].

| | Sensitivity | $B_\mathrm{r}$, $R_\mathrm{r}$, $P_{\mathrm{r},95}$ [µm] | $B_\tau$, $R_\tau$, $P_{\tau,95}$ [] | $F$ [%] | input habit | details |
|---|---|---|---|---|---|---|
| 1 | Consistency check | 0.0, 0.0, 0.0 | 0.0, 0.0, 0.0 | 0.0 | habit mix | $\theta_0, \theta_v, r_\mathrm{eff}, \tau$ in LUT |
| 2 | Interpolation $\mathbf{r_{eff}}$, $\boldsymbol{\tau}$ | 0.4, 3.3, 5.3 | -0.0, 0.5, 0.3 | 5.7 | habit mix | $r_\mathrm{eff}, \tau$ not in LUT |
| 3 | Interpolation $\boldsymbol{\theta_0}$, $\boldsymbol{\theta_v}$ | 0.3, 3.8, 6.8 | 0.0, 0.5, 0.4 | 8.0 | habit mix | $\theta_0,\theta_v,r_\mathrm{eff},\tau$ not in LUT |
| 4a | Habit assumption | -2.4, 10.2, 19.7 | 0.9, 1.4, 3.1 | 72.0 | sol. column | $\theta_0,\theta_v,r_\mathrm{eff},\tau$ not in LUT |
| 4b | Habit assumption | -1.5, 13.1, 29.9 | 0.4, 1.2, 2.5 | 58.1 | hol. column | $\theta_0,\theta_v,r_\mathrm{eff},\tau$ not in LUT |
| 4c | Habit assumption | -4.0, 16.6, 35.1 | 1.4, 2.0, 4.6 | 83.7 | aggregate | $\theta_0,\theta_v,r_\mathrm{eff},\tau$ not in LUT |
| 4d | Habit assumption | 2.3, 11.6, 26.8 | 0.1, 1.3, 2.3 | 51.4 | rosette | $\theta_0,\theta_v,r_\mathrm{eff},\tau$ not in LUT |
| 4e | Habit assumption | 2.2, 13.0, 28.3.6 | -0.7, 2.3, 5.1 | 68.1 | plate | $\theta_0,\theta_v,r_\mathrm{eff},\tau$ not in LUT |
| 4f | Habit assumption | -4.5, 10.5, 26.3 | 0.4, 1.3, 2.8 | 47.7 | droxtal | $\theta_0,\theta_v,r_\mathrm{eff},\tau$ not in LUT |
| 5 | Aerosol concentration | 0.6, 9.0, 18.7 | 0.5, 0.8, 1.1 | 36.0 | habit mix | AOT, $\theta_0,\theta_v,r_\mathrm{eff},\tau$ not in LUT |
| 6 | Instrument noise | 0.3, 3.8, 7.0 | 0.0, 0.5, 0.4 | 8.1 | habit mix | $\theta_0,\theta_v,r_\mathrm{eff},\tau$ not in LUT |
| 7a | +5% Calibration offset | -0.0, 6.0, 12.0 | -0.1, 0.8, 1.4 | 35.6 | habit mix | $\theta_0,\theta_v,r_\mathrm{eff},\tau$ not in LUT |
| 7b | -5% Calibration offset | 0.0, 6.2, 12.6 | 0.3, 0.8, 1.7 | 37.1 | habit mix | $\theta_0,\theta_v,r_\mathrm{eff},\tau$ not in LUT |





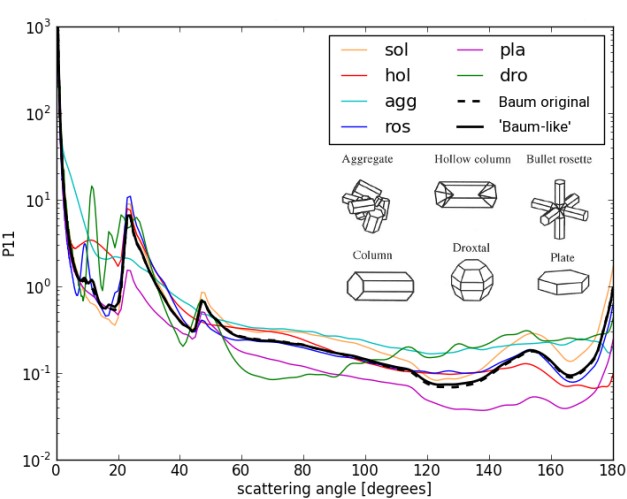

**Figure 1.** Phase functions of individual habits (coloured lines), original Baum et al. (2005) and "Baum-like" reproduced habit mixtures (black lines) for $r_{\mathrm{eff}} = 50\ \mu\mathrm{m}$ and $\lambda = 550\ \mathrm{nm}$. Crystal habits (Yang et al., 2005) shown are droxtals (dro), plates (pla), bullet rosettes (ros), aggregates (agg), solid (sol) and hollow (hol) hexagonal columns.





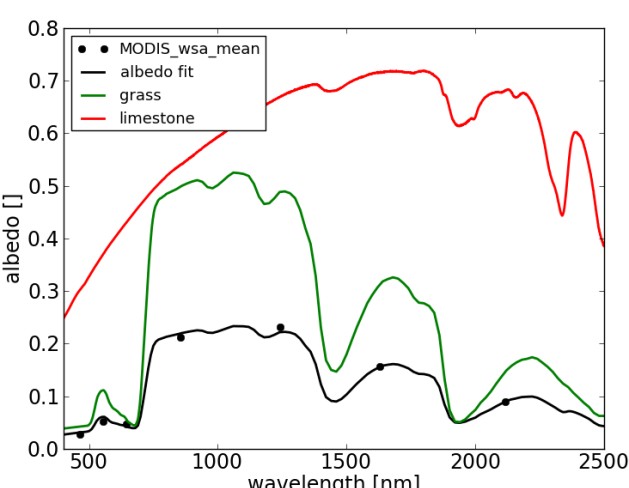

**Figure 2.** Interpolation of MODIS white-sky-albedo (black points) with ASTER spectral albedo of grass (green line) and limestone (red line). Spectral surface albedo for the area around UFS (black line) resulting from the fit $A = 0.39 \cdot A_{\mathrm{grass}} + 0.05 \cdot A_{\mathrm{limestone}}$.



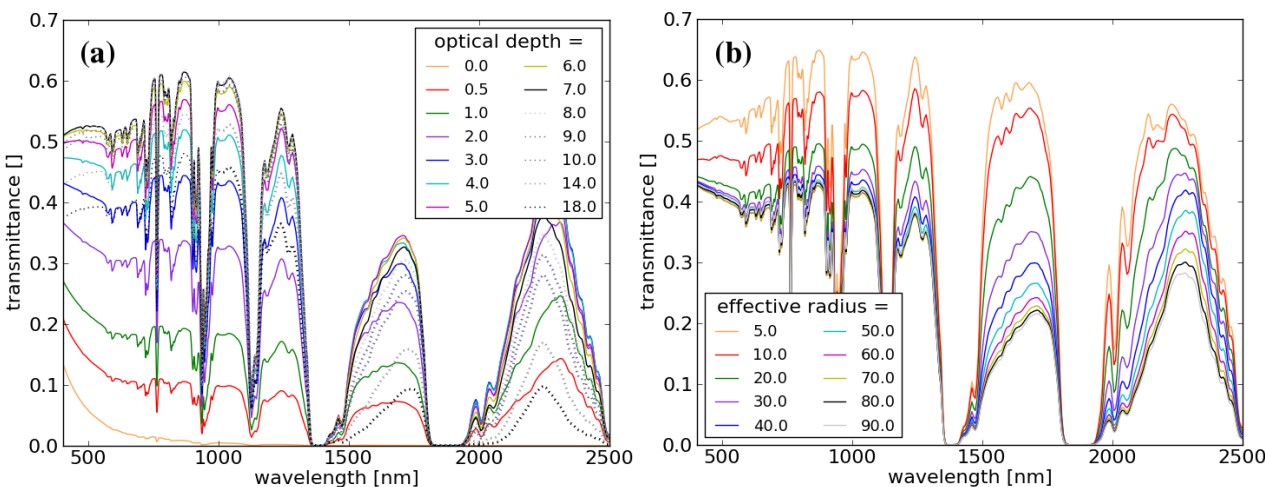

**Figure 3.** Ice cloud transmittance simulations at specMACS spectral sensitivity for $\theta_0 = 36°$, $\theta_v = 0°$, $\phi_{rel} = 180°$, and "Baum-like" habit mixture: Variation of **(a)** optical thickness ($r_{eff} = 40 \, \mu m$) and **(b)** effective radius ($\tau = 3.0$).





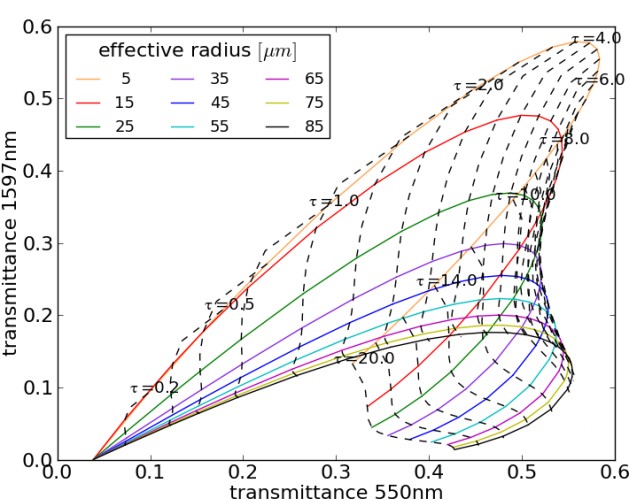

**Figure 4.** Example of a lookup table of transmittance at 550 nm and 1600 nm simulated for $\theta_0 = 36°$, $\theta_v = 0°$, $\phi_{\mathrm{rel}} = 180°$, and habit mixture.




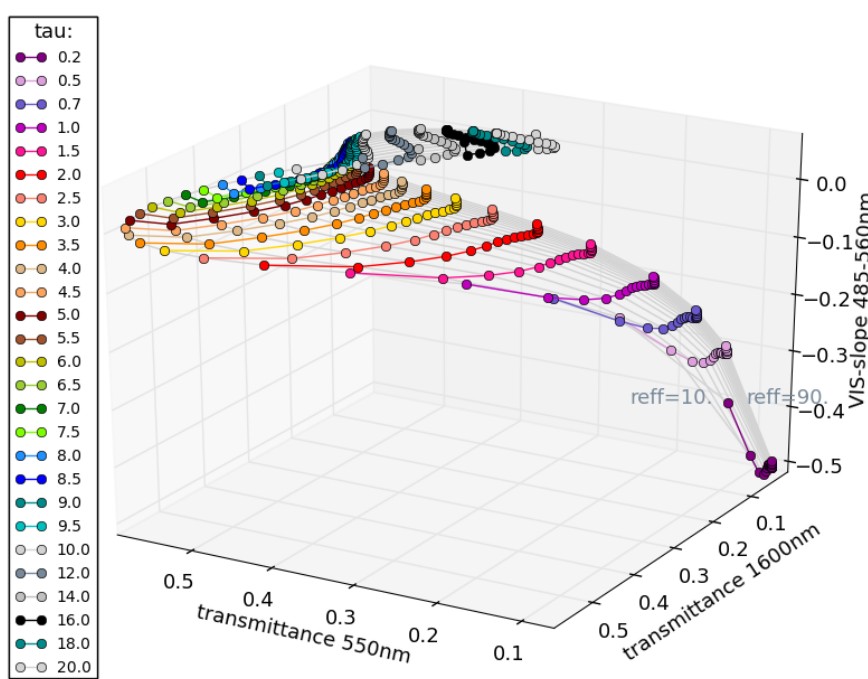

**Figure 5.** 3D lookup table using transmittances at 550 nm, 1600 nm, and the spectral slope $S_{\mathrm{VIS}}$ in the range 485 - 560 nm (simulation for $\theta_0 = 36°$, $\theta_v = 0°$, $\phi_{\mathrm{rel}} = 180°$, and habit mixture).





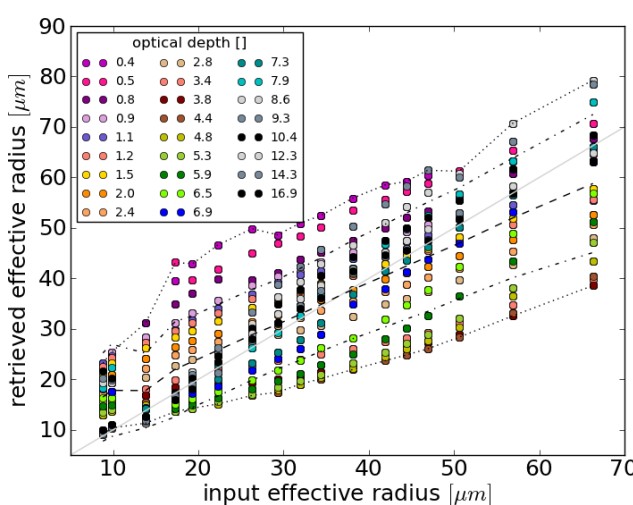

**Figure 6.** Sensitivity to ice crystal habit (test 4d of Table 1): Scatter plot for the effective radius retrieval assuming the habit mixture. Test spectra are simulated for bullet rosettes with $\tau$, $r_{\mathrm{eff}}$, $\theta_0$, and $\theta_v$ not included in the lookup tables. Dashed lines indicate $P_{r,2.5}$ and $P_{r,97.5}$ (according to 95% of all absolute errrors), dash-dotted lines indicate $\pm R_r$, dotted lines the mean for the test cases shown, $\theta_0 = 31.75°$, $\theta_v = 2.25°$, $\phi_{\mathrm{rel}} = 180°$.





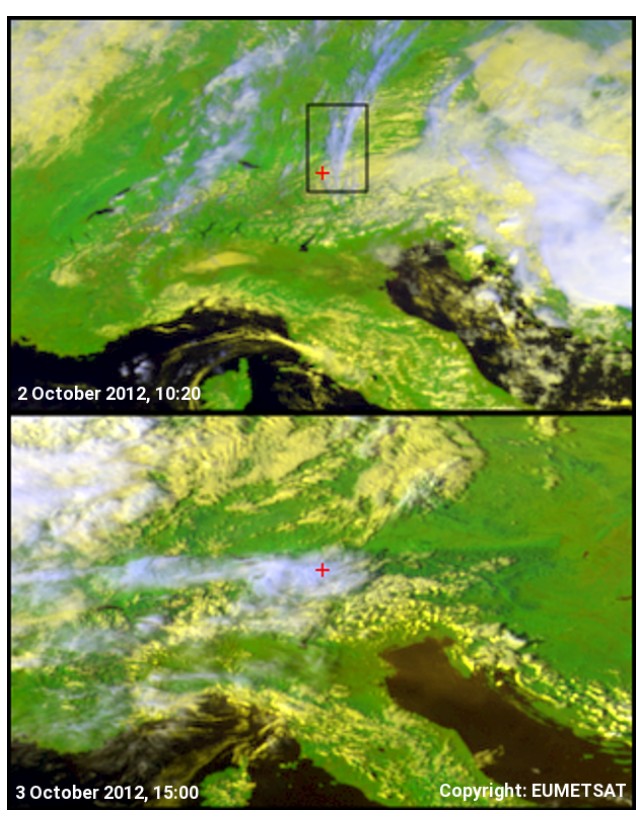

**Figure 7.** Central Europe in a METEOSAT false colour composite (R: 0.6 µm, G: 0.8 µm, B: inverted 10.8 µm): (a) for 2 October 2012, 10:20 UTC and (b) for 3 October 2012, 15:00 UTC. Use of thermal IR information leads to blueish (cold, high clouds) and yellowish (warm, low clouds). The red cross marks the position of UFS at Zugspitze, the black box the approximate location of MODIS data granule in Fig. 11.





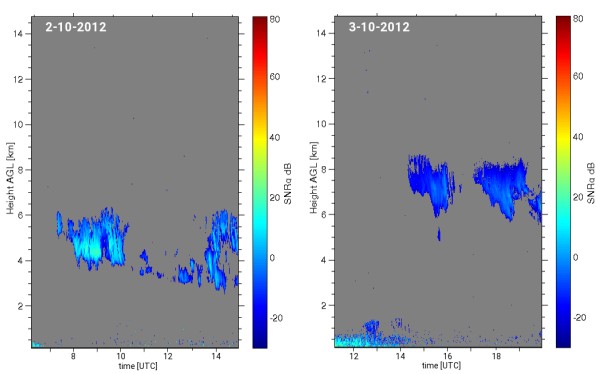

**Figure 8.** Cloud radar cross section from the vertical pointing Ka-band METEK MIRA 35 on UFS Schneefernerhaus (height ASL 2650 m) around time of specMACS measurements.




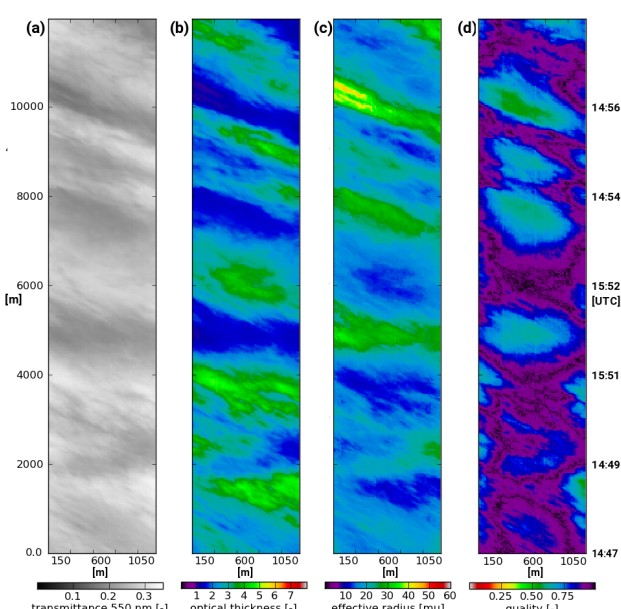

**Figure 9.** specMACS data at 550 nm collected above UFS on 3 October 2012, 14:47-14:57 UTC (a), retrieval of effective radius (b), optical thickness (c) and quality (d) ranging from 0 (no values in LUT within search radius) to 1 (a perfect match in the LUT). The spatial dimensions of the measurement at 2160 x 10800 m are derived from cloud height (see Fig. 8) and advection wind speed.





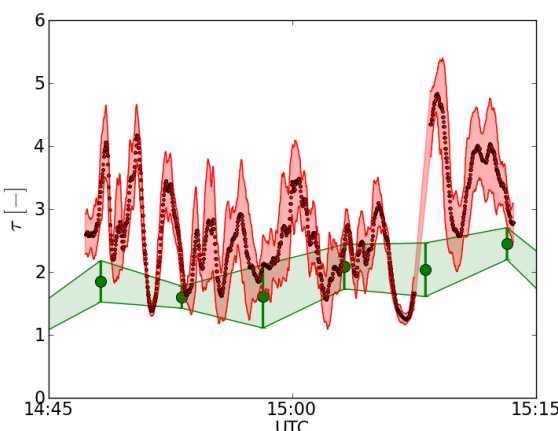 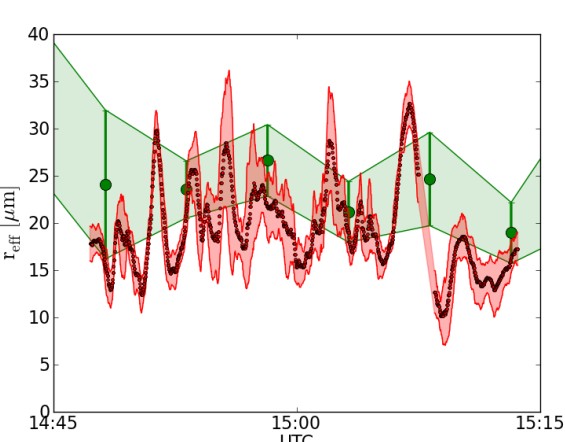

**Figure 10.** Comparison of retrievals for 3 October 2012: red points show 4 second averages of specMACS retrievals for two sequences of data collection 14:47 and 15:13 UTC, larger green dots show METEOSAT-SEVIRI retrievals at the position of UFS. Error bars and green shading for SEVIRI data are related to the 8-connected neighbours, and red shading for specMACS data is related to the standard deviation in all retrievals collected in a 4 second period.





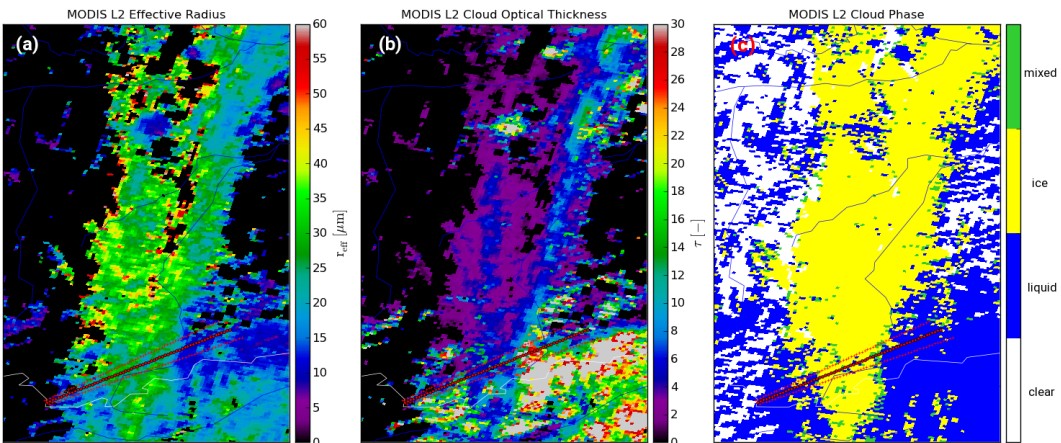

**Figure 11.** MODIS collection 6 effective radius (a), optical thickness (b) and cloud mask (c) for 2 October 2012, 10:20 UTC. Red lines of symbols are related to cross sections in time assuming advection (compare Fig. 13). The first symbol at the lower left end of these lines marks the UFS position.





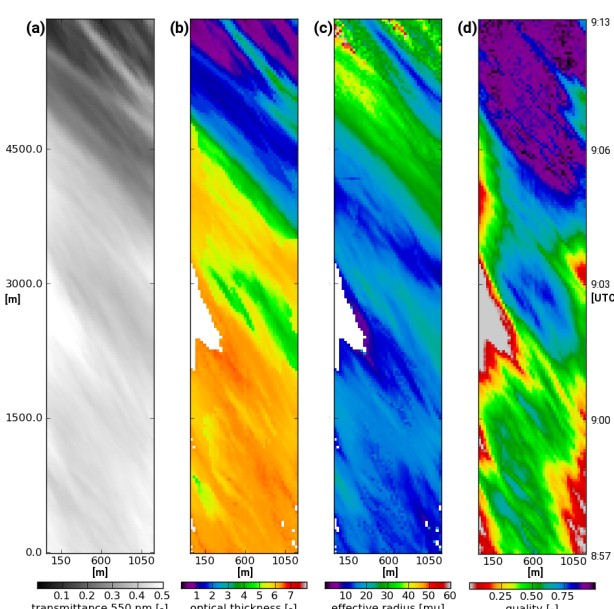

**Figure 12.** specMACS data at 550 nm collected above UFS on 2 October 2012, 8:56-9:10 UTC (a), retrieval of effective radius (b), optical thickness (c) and quality (d) ranging from 0 (no values in LUT within search radius) to 1 (a perfect match in the LUT). The spatial dimensions of the measurement resulting 1200 x 5970 m are derived from cloud height (see Fig. 11) and advection wind speed.

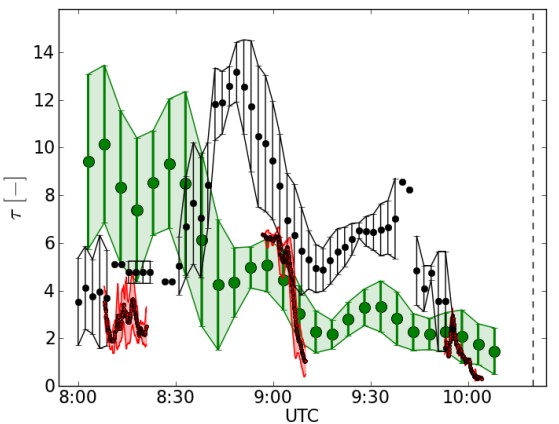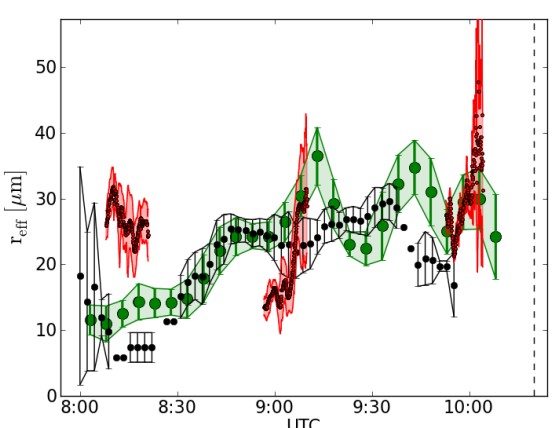

**Figure 13.** Comparison of retrievals for 2 October 2012: black dots show MODIS retrievals for ice clouds along the wind direction line as depicted in Fig. 11, red points show 4 second averages specMACS retrievals for three sequences of data collection 8:08-8:21, 8:57-9:10 and 9:53-10:04 UTC, larger green dots show METEOSAT-SEVIRI retrievals at the position of UFS. Error bars for MODIS retrievals are related to the standard deviation of retrievals in a spatial tolerance region around the wind direction (also compare to Fig. 11), The broken vertical line labels the actual time of the TERRA-MODIS overpass.