# Peer review of "Ground-based imaging remote sensing of ice clouds: uncertainties caused by sensor, method and atmosphere"

_Atmospheric Measurement Techniques, 2016_

## Referee Comment (RC1) · Anonymous Referee #2 · 13 May 2016

**Review of: "Ground-based imaging remote sensing of ice clouds: uncertainties caused by sensor, method and atmosphere" by Zinner et al.**

**1   General remarks**

The manuscript provides an in-depth description and sensitivity study of ground based cirrus retrieval using transmitted solar radiance. Extensive radiative transfer simulations are used to present the retrieval approach and quantify retrieval uncertainties of

different sources. A transmittance ratio in the visible part of the solar spectrum is used to overcome the ambiguity between transmissivity and cloud optical thickness. The method is applied to measurements of the hyperspectral line imager specMACS for two case studies. The results are compared with satellite retrieval of cloud properties.

The application of cloud transmissivity retrieval to spectral solar imager observations is a potential tool for future investigation of clouds with high spatial resolution. Especially high resolution fields of cloud particle size may help to understand microphysical processes in cirrus. In this regard, the manuscript provides an important contribution to current and future research and is worth to be published.

However, in my opinion the manuscript lacks of three major issues which have to be reassessed in detail before publishing the manuscript. First, existing similar methods have not been discussed appropriately in the manuscript. Second the authors did not apply the latest improvements of transmittance retrieval but rather stick to a classical approach. Additionally, the second case study was an unfortunate choice as the comparison with satellite observations is not meaningful due to contamination by low level clouds.

Below, I compiled a list of comments which have to be considered in a revised version of the paper. There might be some contradictory statements resulting from my misinterpretation of the text when first reading. I am sure the authors will know how to weight in such cases and how to improve the text to avoid misinterpretations by other readers.

**2  Major comments**

**Existing methods have not been considered**

In the manuscript some available studies have not been discussed properly. E.g., the problem discriminating the ambiguity of thin and thick cirrus clouds was already solved

by Brückner et al. (2014) who applied a similar retrieval for transmissivity measurements and also used a ratio in visible wavelength for a third coordinate in the retrieval grid separating thin and thick cirrus.

Similarly, in the sensitivity study and in the conclusions shape effects for transmissivity are discussed. Results of the sensitivity study should by compared to Schäfer et al. (2013) who did similar sensitivity studies for the retrieval of optical thickness by transmissivity in case of tropical cirrus. Additionally, Schäfer et al. (2013) present an approach to estimate ice crystal shape. This could be applied to some extend for the measurements of specMACS as well reducing the retrieval uncertainty due to the assumption of ice crystal shape.

**State of art transmissivity retrievals**

I wounder, why the authors do not build their retrieval algorithm on the existing improvements introduced by Brückner et al. (2014), McBRide et al. (2011), LeBlanc et al. (2015) for transmissivity retrievals? These retrieval are based on radiance ratios instead of absolute radiance/transmittance and do improve the retrieval uncertainty. This was even discussed by the authors in the conclusions. Therefore, I wounder why the authors did not apply these new methods despite knowing that they provide more precise results than the "classical" Nakajima-King approach. As all look-up tables seem to be calculated for the full spectral range given by specMACS, switching to ratios should be an easy task.

**Case study 2. October**

The choice of the second case study presented in the manuscript was rather unfortunate. As discussed in detail by the authors, the comparison between satellite and ground-based measurements suffers due to low level clouds contaminating the cloud retrieval of the satellite instruments. Therefore, the data sets are in general not comparable in my point of view. Presenting this comparison is not meaningful and does not add any value to the manuscript. At least not for the main subject, cirrus transmittance retrieval using imaging spectrometers. So I would suggest to choose another case or at least remove the satellite comparison. Instead it might be worth to compare and discuss the differences in cloud properties and retrieval uncertainties between both cases.

**3  Minor comments**

**P1, 7:** Typo: "noise,."

**P1, 12:** "verification": In what way was this accomplished? Comparison with satellite data?

**P1, 15:** "variability": variability of what? clouds?

**P2, 10:** Greenhouse effect of cirrus also depends on temperature/altitude of the cirrus!

**P2, 32:** Airborne remote sensing might be suited better for improving satellite observations as these have at least a similar geometry. For motivation, ground-based techniques can also be considered for long term monitoring operations, e.g., ARM sites, CloudNet.

**P4, 9:** How many spatial pixel does one line have?

**P5, 31:** Typo or double negation? "does not show an unambiguous relationship" change to "does show an ambiguous relationship"

**P6, 8:** Emphasize that this method is well known. Not only for existing transmittance retrieval as discussed above, but also any cloud mask/cloud fraction algorithms in all sky imaging uses such ratios.

[Figure]

**P6, Section 3.2:** Section 3.2. somehow does not fit in the outline of the manuscript. It should be placed at 2.2 where already model, surface albedo and ice crystal scattering properties are introduced.

**P7, 31:** What about particle size? This should also affect the thresholds.

**P8, 11:** wording: "...unsing with respect..." please change.

**P9, Section 3.5:** The section is quite long and needs to be split into single paragraphs to improve the readability.

**P12, 25:** Give numbers for the mean values.

**Figure 3:** Indicate the wavelengths selected for the retrieval.

**Fig 7 and 8:** I would suggest to merge Fig. 7 and 8 to reduce the number of figures.

**Fig 8,9,12:** Increase font size of axes and labels.

**Bibliography**

Brückner, M., Pospichal, B., Macke, A., and Wendisch, M.: A new multispectral cloud retrieval method for ship-based solar transmissivity measurements, J. Geophys. Res. Atmos., 119, 11.338–11.354, doi:10.1002/2014JD021775, 2014.

LeBlanc, S. E., Pilewskie, P., Schmidt, K. S., and Coddington, O.: A spectral method for discriminating thermodynamic phase and retrieving cloud optical thickness and effective radius using transmitted solar radiance spectra, Atmos. Meas. Tech., 8, 1361–1383, doi:10.5194/amt-8-1361-2015, 2015.

McBride, P. J., Schmidt, K. S., Pilewskie, P., Kittelman, A. S., and Wolfe, D. E.:

A spectral method for retrieving cloud optical thickness and effective radius from surface-based transmittance measurements, Atmos. Chem. Phys., 11, 7235–7252, doi:10.5194/acp-11-7235-2011, 2011.

Schäfer, M., Bierwirth, E., Ehrlich, A., Heyner, F., and Wendisch, M.: Retrieval of cirrus optical thickness and assessment of ice crystal shape from ground-based imaging spectrometry, Atmos. Meas. Tech., 6, 1855–1868, doi:10.5194/amt-6-1855-2013, 2013.
* * *

---

## Referee Comment (RC2) · Anonymous Referee #1 · 24 May 2016

Interactive comments on "Ground-based passive remote sensing of thin ice clouds: challenges caused by sensor, method and atmosphere" by Tobias Zinner et al.

**General Vote**

The manuscript provides an important contribution to the state of the art focused on remote sensing of cirrus. I recommend its publication after the authors have revised the manuscript with regard to the comments listed below.

**Referee´s Synopsis**

The authors introduce a method to retrieve cloud optical (cloud optical thickness $\tau$) and microphysical (effective particle size $r_{eff}$) properties from ground-based measurements of solar spectral radiance transmitted through cirrus. To avoid ambiguity in the retrieved data the authors extend the classical approach by Nakajima and King (1990) by a third dimension; a slope fit between 485 and 560 nm. To test this approach and to estimate the retrieval uncertainties, the manuscript provides an intensive sensitive study on different uncertainty sources using extensive radiative transfer simulations. Furthermore, the retrieval results for two test cases (measured in the visible and near-infrared wavelength range using the imaging spectrometer system specMACS during the ACRIDICON-Zugspitze campaign) are compared to satellite retrievals of cloud optical thickness and effective radius.

**Major comments**

1. **Literature review:** One main issue of the manuscript is the insufficient literature review and comparison to recently published studies. This concerns in particular the handling of the ambiguity between transmitted solar spectral radiance and cloud optical thickness as well as to comparisons of the results of the sensitivity study to literature values.

   a. **Ambiguity:** In the current manuscript, a third dimension is applied to the classical two-wavelength cloud retrieval by Nakajima and King (1990) to avoid the ambiguity between transmitted solar spectral radiance and cloud optical thickness. This third dimension is given by a slope-fit/ratio in the visible wavelength range between 485 and 560 nm. Recently, Brückner et al. (2014) published a similar method using ratios in the visible wavelength range for the third dimension. The method presented by Brückner et al. (2014) definitely should be considered and discussed in the current manuscript.

   b. **Sensitivity study:** The present manuscript provides a detailed and impressive sensitive study on possible retrieval uncertainties. However, the results should be compared to the results of the sensitivity study given by Schäfer et al. (2013). For a cirrus retrieval adapted to the measurements with a ground-based imaging spectrometer in the visible wavelength range, Schäfer et al. (2013) investigated the retrieval uncertainties of cloud optical thickness retrievals, e. g. including surface albedo and cirrus crystal shape.

2. **2nd test case:** From my point of view, the discussion of the second test case from 2 October 2012 should be removed from the manuscript. The first case is already sufficient to demonstrate the ability of the introduced cirrus retrieval to give proper results. The manuscript will not benefit from the second case. Of course, it would be nice to have two satellite products to compare, but due to the contamination by low clouds, a comparison will not be significant. Furthermore, the data seem to be overexposed at multiple parts of the image, which may be the reason that no cloud retrieval could be adapted at those parts.

**Minor and technical comments**

1. **Acronyms**: Acronyms are often used several times before they were introduced the first time. Examples are LMU, specMACS, ACRIDICON, MODIS, SEVIRI, CloudSat, CALIPSO. I don't know if I got them all. Please check all acronyms throughout the manuscript and introduce their full names whenever they are used for the first time.

2. **Indices and units:** Indices and units are sometimes written in italic letters and sometimes in non-italic letters. Throughout the manuscript this happens also for one and the same index or unit. For reasons of consistency you should write all indices and units in non-italic letters.

3. **P1, L7:** Typo, remove dot from "noise,."

4. **P3, L7:** Typo: persepctive → perspective

5. **P5, L29:** Remove "-" from "lookup-table". Throughout the whole text "lookup table" is written without hyphen (-)

6. **P8,L11:** remove "using"

7. **P9, L13:** lookup → lookup table

8. **P12, L12:** "At quality values above 0.5." is no full sentence. You could connect this one to the sentence before.

9. **P11, L10-L14:** The phase discrimination is the first step of the retrieval procedure. Therefore, my recommendation is to shift the discussion of this part to the beginning of chapter 3.5 instead of keeping it at the end.

10. **Fig. 6:** Some colors are used twice. For example for tau=10.4 and tau=16.9, for tau=8.6 and tau=12.3, … Please revise this figure.

11. **Fig. 8, 9, 12:** Please increase font size

12. **Fig. 8:** Only as a suggestion. Would it be possible to indicate the time of measurement in Fig. 8?

13. **Fig. 12:** Time axis does not fit the time given in the figure caption and does not fit the time given on P11, L18

**Bibliography:**

Brückner, M., Pospichal, B., Macke, A., and Wendisch, M.: A new multispectral cloud retrieval method for ship-based solar transmissivity measurements, J. Geophys. Res. Atmos., 119, 11.338–11.354, doi:10.1002/2014JD021775, 2014.

Schäfer, M., Bierwirth, E., Ehrlich, A., Heyner, F., and Wendisch, M.: Retrieval of cirrus optical thickness and assessment of ice crystal shape from ground-based imaging spectrometry, Atmos. Meas. Tech., 6, 1855–1868, doi:10.5194/amt-6-1855-2013, 2013.

---

## Referee Comment (RC3) · Anonymous Referee #3 · 26 May 2016

This is another in a series of valuable and interesting papers from the community that demonstrate the utility of spectral information for retrieving properties of clouds from passive shortwave remote sensing. It focuses specifically on cirrus clouds based on ground-based spectral imaging. The challenge for this type of cloud is the small optical thickness combined with a high degree of spatial variability, as well as the unknown composition in terms of crystal shape and roughness. Transmittance measurements of liquid and ice clouds alike also suffer from the ambiguity of the relationship between optical thickness and downwelling radiance, where an increasing number of cloud particles initially increases the amount of diffuse radiation that is scattered out of the direct beam and towards the sensor. At larger optical thickness (generally above four), attenuation decreases the downwelling radiation below clouds. The resulting ambiguity of a low- and a high-optical thickness retrieval for one radiance observation was tackled through several avenues (thorough literature reviews are provided by *McBride et al.* (2011) and *Brückner et al.* (2014); this paper is a bit sparse in this regard; a reference list is given at the end of this review). The solution to the ambiguity problem was found by *LeBlanc et al.* (2015) by introducing the spectral slope around 550 nm, but neither this nor a similar paper by *Brückner et al.* (2014) were based on an imaging spectrometer as done in the current manuscript. Figure 6 and the accompanying text from *LeBlanc et al.* (2015) are included below for clarification as the genesis and physics are not very well described by the manuscript in its current form.

[Figure]

**Figure 6. (a)** Normalized transmitted radiance of liquid water clouds of varying optical thickness and effective radius of 20 μm and **(b)** the slope of normalized radiance for the wavelength range of 530–610 nm, $\eta_{11}$, as a function of $\tau$ for ice and liquid clouds, evaluated for three different effective radii. In **(a)**, the slope in the visible is identified by the highlighted region. The normalized radiance spectra and the $\eta_{11}$ calculated from them were modeled with ancillary inputs based on 25 May 2012 for a solar zenith angle of 50°.

Quote from LeBlanc et al. (AMT, 2015): For clouds with $\tau < 4$, where radiance in the mid-visible is still increasing with $\tau$, the normalized transmitted radiance spectra show an influence from molecular scattering. The spectra in Fig. 3a for $\tau = 0.2$ matches more closely the clear sky spectra, which is inversely proportional to the fourth power of the wavelength, than the normalized radiance spectra for $\tau = 100$, which is roughly proportional to the inverse of the wavelength. As $\tau$ is reduced, the magnitude of signal at wavelengths between 550 and 700 nm decreases and its slope becomes more negative until they match the spectrum of clear sky. The clear sky spectrum (green spectrum in Fig. 3a) is entirely dependent on scattering by molecules (Rayleigh scattering) and the solar zenith angle. The slope of the spectrum in the visible is proportional to $\tau$ until scattering by cloud particles dominates scattering by molecules. This transition occurs at lower $\tau$ for ice clouds (near 1) than liquid clouds (near 2), obscured by radiance

transmitted through optically thicker clouds in Fig. 3. After this transition, the slope of normalized radiance in the visible varies less and depends on $\tau$, re, and $\phi$, rather than on molecular scattering. Similar results are also observed by Brückner et al. (2014), where instead of a slope in the mid-visible, they used a ratio of transmittance at 450 and 680 nm.

The other noteworthy paper needed to put this manuscript into context is the one by *Schäfer et al.* (2013), which sought to capitalize on *imaging* (i.e., radiance measurements as a function of viewing angle) to derive optical thickness and habit information from transmitted radiance.

Since the manuscript emphasizes (p3, l16-19) that it combines spectral information with imaging for the solution of the ambiguity problem mentioned above, it should discuss *Brückner et al.* (2014), *LeBlanc et al.* (2015), and *Schäfer et al.* (2013) properly (*Brückner et al.* (2014) is currently not cited). At the very least, it should briefly describe how these three papers resolve the ambiguity. This would show that the current manuscript actually pursues the same method as introduced by *LeBlanc et al.* (2015) (not only by using the same spectral slope, but also by normalizing the radiance spectrum first) with the exception that the current manuscript uses *fewer* spectral parameters than the LeBlanc paper; on the other hand, the *imaging* capabilities that the *Schäfer et al.* (2013) manuscript relies on are not used to resolve the ambiguity; instead they serve to provide context measurements. For these reasons, the statement "We will present a combination of both, a solution for the transmittance ambiguity…" should be revised to reflect that the same or a similar approach as employed by the *LeBlanc et al.* (2015) and *Brückner et al.* (2014) papers was used here. Instead of making this clarification at this point, it may be more appropriate in Section 3.1 where the idea is introduced (either way, the proper credit to the origin of the idea should be given). In the same vein, the statement "a new method for the retrieval of ice cloud optical thickness … from transmitted radiance" in the summary is probably not quite correct, since the additional observable (spectral slope in the mid-visible) had been proposed earlier.

Notwithstanding the minor concern about the origin and genesis of the idea, the manuscript provides a valuable new perspective on the challenges that are encountered when retrieving cirrus cloud optical properties (as mentioned at the beginning of this review). The direct inter-comparison of ground-based retrievals with concurrent satellite observations (geostationary and polar-orbiting) seems especially valuable, revealing that there may be significant differences even from satellite to satellite retrieval that need to be resolved in the future. The main suspect for the discrepancies between the different algorithms are the insufficiently constrained scattering phase functions for cirrus clouds. Spectral imaging is probably the method of choice for resolving this problem in the future, and the manuscript is a nice step towards addressing it. The sequential comments below outline some doubts about the methodology on how the phase function (crystal shape) related biases are estimated, but those, as well as all the other suggested changes should be fairly easy to implement in the revised manuscript.

Sequential major comments:

P2, L27: King et al. (2004) is the wrong reference for the adaptation of the Nakajima/King algorithm to ice clouds. At this point, the MODIS retrievals were already fully operational. One of the Platnick references is probably more appropriate; the cited paper is for the retrievals over bright surfaces, an entirely different problem. A nice review of ice cloud retrievals (with emphasis on the problematic phase function) is provided by *Wang et al.* (2014).

P3, L16: Clarify (here or later) that *LeBlanc et al.* (2015) used the spectral slope of normalised mid-visible transmittance (or radiance), and that this is what the algorithm in this manuscript relies on as well (rather than using the *combination* of the spectral slope and imaging capabilities, as the subsequent sentence implies). See also the comments on pages 1 and 2 of this review. Also change this appropriately at later occurrences (for example in the summary); it should be stated that the idea for resolving the ambiguity problem was first introduced by the 2014 and 2015 papers.

P4, L24: 16 streams seem too few to properly resolve the features of the scattering phase function unless the cited intensity correction works properly for small and medium optical thickness values (single or low-order scattering) – this is of course especially true for scattering angles near the halo angles or other pronounced features. It would be highly recommended to show a plot of simulated downwelling radiance as a function of viewing angle such that the ability of the RT code to simulate the halo in the right place becomes credible. Too few streams in the solver may misplace the halo despite the application of the intensity correction. Perhaps such a plot could be added in an appendix?

Section 2.2.2, optical properties: Using all six habits from "HEY" may not be appropriate since some of the habits are highly unlikely to occur in the study region. The authors do mention that the errors due to crystal shape represent an upper bound for this reason, but even that may not be correct if a certain habit does not occur at all in the kind of cloud that is observed. This point, combined with the previous one (are the features of the phase functions are resolved by the intensity-corrected 16-stream radiances?) cast some doubt on the concluded magnitude of the bias due to unknown habit.

Section 2.2.3, surface albedo: MODIS surface albedo products are known to have problems in regions with pronounced topography. Given that, why was the sensitivity to surface albedo and its uncertainty not tested in similar ways as done for the other parameters (see table 1)? This seems even more important than testing the sensitivity to the presence of aerosols.

P6, Eqn (2) This parameter is equivalent to $\eta_{11}$ from *LeBlanc et al.* (2015), except that they used radiance (not transmittance) to derive it, which is conceptually the same – see also previous comments.

P6,l24: "Consequently, it is *now* possible to..." See comments above (origin/genesis).

P7,l10: Why does the LUT only include three different relative azimuth angles? Can this be justified by the geometric arrangement and a special sun-viewing geometry? The relevant angle for near-single scattering remote sensing (i.e., up to about 1 optical thickness) should be the scattering angle, not the viewing zenith + azimuth angle, correct? This may only be a minor point – however, typical LUTs usually feature more than the azimuth angles used in this study, so the question is why this study can get by with so few. A plot that shows the dependence of the radiance on viewing azimuth angle for a small optical thickness would help giving some credence to the approach. Combined with an earlier suggestion: The approach can only be regarded as sound if the halo (or another feature) can be simulated in the right place (in the 2D image) and resolved both in viewing zenith and azimuth angle.

P8,Eqn 3: It is unclear why the three terms (T550, T1600, SVIS) are each given equal weight in determining dT. After all, the measurement uncertainties propagate into each of these in different ways, and the individual terms should be weighed by their (different) uncertainty, rather than applying a global weight for the different LUT points as proposed in Eqn 5 later on.

P8/P9: Related to the comment above: How is the uncertainty in COD and REFF derived from the uncertainty in the measurements (the radiances)? dT cannot be regarded as a metric for the uncertainty because it is a merely technical quantity which would change for a different LUT gridding. Also, trading standard deviation for uncertainty as done on p12,l20-21 does not help. How do we know for certain that the error does not exceed the standard deviation? Particularly for small crystal size (on the order of 10-20 micron) and the optical thickness range shown, the effective radius may be highly uncertain because of the issues described by *McBride et al.* (2011). An error propagation analysis from the radiances to the retrieval parameters should be performed, not in terms of the LUT gridding, but in terms of the underlying physics. This is probably the most work-intensive revision that is suggested in this review.

P10,L9: See earlier comment concerning the azimuth vs. scattering angle. Azimuth does not seem to be the appropriate parameter to consider for singe-scattering conditions.

P10,l22-23: Would it help to limit to such habits that can realistically occur for the type of cloud under consideration? What about the distinction between habit and surface roughness (which seems to have a larger role than the habit when it comes to smooth versus "featured" phase functions)? Is it possible to say something about the roughness using the imaging capabilities of the instrument (in the vein of the Schäfer paper)?

P12,l20-21: See earlier comment: The standard deviation cannot replace a physics-based error analysis. Also: The impact of surface albedo uncertainty should be part of the retrieval error, unless it is deemed to be negligible.

P14,l16 & l18-20: Please consider earlier comments regarding the origin of the "*new third parameter*" for the removal of the ambiguity.

P16,l1-5: I commend the authors on tackling the arduous task to reduce the retrieval uncertainty due to crystal shape using spectral imaging!

Sequential minor comments (language/typos):
P1,l1: "for retrieval of" → "for *the* retrieval of" ?
P1,l7: delete "unknown" ?
P1,l7: typo with "." and "," at end of line
P1,l10: "be caused" → "arise" ? (sounds awkward otherwise)
P1,l11: "For optical…" Sentence does not make sense, perhaps because of the first "are"?
P4,l19: "We have used" → "We used" ?
P6,l27: "needed" → "required" (sounds a bit less awkward)
P10,l8: "on illumination" → "on other illumination" ?
P11,l16: duplication of "presented"

References:

Brückner, M., B. Pospichal, A. Macke, and M. Wendisch (2014), A new multispectral cloud retrieval method for ship-based solar transmissivity measurements, J. Geophys. Res. Atmos., 119, 11,338–11,354, doi:10.1002/2014JD021775.

LeBlanc, S. E., Pilewskie, P., Schmidt, K. S., and Coddington, O.: A spectral method for discriminating thermodynamic phase and retrieving cloud optical thickness and effective radius using transmitted solar radiance spectra, Atmos. Meas. Tech., 8, 1361-1383, doi:10.5194/amt-8-1361-2015, 2015.

McBride, P. J., K. S. Schmidt, P. Pilewskie, A. S. Kittelman, and D. E. Wolfe (2011), A spectral method for retrieving cloud optical thickness and effec- tive radius from surface-based transmittance measurements, Atmos. Chem. Phys., 11(14), 7235–7252, doi:10.5194/acp-11-7235-2011.

Schäfer, M., Bierwirth, E., Ehrlich, A., Heyner, F., and Wendisch, M.: Retrieval of cirrus optical thickness and assessment of ice crystal shape from ground-based imaging spectrometry, Atmos. Meas. Tech., 6, 1855-1868, doi:10.5194/amt-6-1855-2013, 2013.

Wang, C., P. Yang, A. Dessler, B. A. Baum, and Y.-X. Hu, 2014: Estimation of the cirrus cloud scattering phase function from satellite observations *J. Quant. Spectrosc. Radiat. Transfer, 138, 36-49*.

---

## Author Comment (AC1) · 8 Aug 2016

Dear Reviewer,

thank you for your support and for your suggestions for improving our manuscript.

In general, all reviewers suggest to strengthen the literature review, especially to improve the discussion of earlier publications on imaging ice cloud remote sensing (Schaefer et al. 2013) and the introduction of the visible spectral slope solution for the transmissivity ambiguity (Brueckner el al. 2014 and Le Blanc et al 2015). This is an obvious weakness of our manuscript. The reason for this negligence on our side is partly due to the fact that our manuscript has had a long history already. In our group the spectral slope approach originally goes back to a Master's thesis of co-author Petra Hausmann from 2012. We obviously noticed that "our approach" was published meanwhile in proper journals by others. Even though this is no excuse for gaps in our literature review, it might explain why we do not want to state any direct "use" or "application" of ideas introduced by the aforementioned authors. In our revision we do both, we try to strengthen our literature discussion, and at the same time we would like to include the Hausmann Master's thesis from 2012 as a reference. Although it is no peer-reviewed publication it is an official university thesis in English language available online.

Point by point reply to all major comments (all minor were considered as suggested, apart from the one mentioned below):

**2 Major comments**

**Existing methods have not been considered**

In the manuscript some available studies have not been discussed properly. E.g., the problem discriminating the ambiguity of thin and thick cirrus clouds was already solved by Brückner et al. (2014) who applied a similar retrieval for transmissivity measurements and also used a ratio in visible wavelength for a third coordinate in the retrieval grid separating thin and thick cirrus.

Similarly, in the sensitivity study and in the conclusions shape effects for transmissivity are discussed. Results of the sensitivity study should by compared to Schäfer et al. (2013) who did similar sensitivity studies for the retrieval of optical thickness by transmissivity in case of tropical cirrus. Additionally, Schäfer et al. (2013) present an approach to estimate ice crystal shape. This could be applied to some extend for the measurements of specMACS as well reducing the retrieval uncertainty due to the assumption of ice crystal shape.

> => The section in the introduction now reads:
>
> ```
> Recently Brückner et al. (2014) as well as LeBlanc et al. (2015)
> presented similar solutions for unambiguous retrievals of optical
> thickness and effective radius for pointing system without
> providing imagery. Both suggest the use of spectral slopes in the
> visible to separate between the two optical thickness regimes.
> We will present a combination of both, a solution for the
> transmittance ambiguity using a similar spectral slope (following
> ideas of Hausmann, 2012) and results for imaging measurements which
> provide context information on the distribution of optical
> thickness and effective radius over a large area.
> ```

We also mention Brückner and LeBlanc at the end of section 3 "Retrieval..." where we presented our version of the idea and in the section 5 "Summary and Discussion".

We included a comparison to the Schäfer et al results into the discussion section:

Schäfer et al. (2013) also assessed the sensitivity of their ground-based cirrus optical thickness retrieval to variation of certain parameters. The values can not be directly compared to our results, as they only refer to a small number of specific situations regarding observation geometry and cirrus situation and not a large range of combinations as in our sensitivity test. For variation of crystal habit and for small optical thickness up to 1 they showed large relative differences up to 80% with average absolute differences at 0.1. Though such cases are contained in the sensitivity test shown here, average impact over many different situations is smaller. Schäfer et al. (2013) also present large uncertainties for an albedo variation. This is caused by their choice of a test albedo which is extremely different from the measurement situation, while here it was assumed that the general albedo situation can be characterized well and remaining uncertainty has only small impact.

We also extended the final discussion of the possibilities to exploit the spatial distribution of transmissivity similar to Schäfer et al.:

Of course the most important step forward would consist in a reduction of the crystal type uncertainty. The halo regions around 22° and 46° scattering angle were avoided here for our spectral approach. Uncertainties can be expected to be higher in these regions with strong angular gradients of transmittance under single scattering conditions, if no additional information on crystal habits is available. However the imaging capabilities of the specMACS sensor (especially if combined with a scanning platform, see Ewald et al., 2015) do not only allow to successfully avoid these regions for the spectral evaluation, but would allow for the utilization of the spatial distribution of transmittance in these regions to provide the missing information. Use of this spatial distribution could provide important constraints regarding the present average phase function as Schaefer et al. (2013) demonstrated. Especially the presence of optical scattering phenomena like type and intensity of halo displays could be used to identify specific particle shapes and orientation and information on the mixture with less perfect rough ice particles. A combination of the presented method with additional information of this kind will be the next step in our effort to provide better ice cloud property observations.

**State of art transmissivity retrievals**

I wounder, why the authors do not build their retrieval algorithm on the existing improvements introduced by Brückner et al. (2014), McBRide et al. (2011), LeBlanc et al. (2015) for transmissivity retrievals? These retrieval are based on radiance ratios instead of absolute radiance/transmittance and do improve the retrieval uncertainty. This was even discussed by the authors in the conclusions. Therefore, I wounder why the authors did not apply these new methods despite knowing that they provide more precise results than the "classical" Nakajima-King approach. As all look-up tables seem to be calculated for the full spectral range given by specMACS, switching to ratios should be an easy task.

> => In part the answer was given in the beginning of our response. That means development of the described techniques was work going on in parallel to the above mentioned not afterwards. The development of a retrieval purely based on ratios would be a totally new effort not within the scope of the project that lead to our manuscript. Apart from this general organizational problem, lookup tables were only generated for the parts of the spectrum needed for our retrieval. This would therefore be related to extensive reprocessing of simulations.

**Case study 2. October**

The choice of the second case study presented in the manuscript was rather unfortunate. As discussed in detail by the authors, the comparison between satellite and ground-based measurements suffers due to low level clouds contaminating the cloud retrieval of the satellite instruments. Therefore, the data sets are in general not comparable in my point of view. Presenting this comparison is not meaningful and does not add any value to the manuscript. At least not for the main subject, cirrus transmittance retrieval using imaging spectrometers. So I would suggest to choose another case or at least remove the satellite comparison. Instead it might be worth to compare and discuss the differences in cloud properties and retrieval uncertainties between both cases.

=> We think the second test case should stay in the manuscript.

> (1) We do not only want to show a single perfect example, but also show an example where the quality is not so good for good reasons ("quality" was renamed "significance" following a comment from another reviewer).

> (2) This second example is also interesting because it demonstrates the possible advantages of a ground-based method. Looking upward, clouds below the mountain top do not directly affect the retrieval, except that they increase the albedo (in contrast to the satellite retrievals which obviously are affected). Very likely our results are the best possibility to provide a "ground truth" for cirrus satellite retrievals in such situations. The possible implications of albedo changes by the underlying cloud patches around the sensor position are also discussed in an additional "spectral albedo" test case in the sensitivity tests and mentioned for this example. We discuss that in the end of this section:

> An interesting aspect of this complex example is the demonstration
> of the potential of a ground-based method to provide accurate cloud
> properties compared to satellite methods, especially for thin
> cirrus. The same quantities are retrieved by both methods,
> utilising similar wavelength bands, but the ground-based method
> benefits from its much higher spatial resolution which allows to
> separate different parts (or layers) of the observed cloudiness. In
> the ground-based data there might still be an impact of increased

albedo (low level cumulus below the instrument). The low levels of
significance of our results at larger sensor zenith angles might be
a sign of it (see Fig. 12d). Nonetheless the ground-based method is
less affected by this problem and generally most likely much better
at retrieving thin ice cloud properties than the satellite methods.

**3  Minor comments**

**P6, Section 3.2:** Section 3.2. somehow does not fit in the outline of the manuscript. It
should be placed at 2.2 where already model, surface albedo and ice crystal scattering
properties are introduced.

> => In section 2 we describe all tools that were available to us when we started and which
> everybody else could use. Section 3 describes the new method. I'm not sure whether it is
> useful to renumber section 3.1/3.2/3.3 to 2.3.1/2.3.2/2.3.3 ?

**Fig 8,9,12:** Increase font size of axes and labels.

> => I increased font size of Fig 8. For Figure 9 and 12 I would prefer a larger image size
> which also will depend on the later layout.

**Reference:**

- Hausmann, P.: Ground-based remote sensing of optically thin ice clouds, 89 pages,
  Master's thesis, Ludwig-Maximilians-Universität, Munich, http://www.meteo.physik.uni-
  muenchen.de/DokuWiki/lib/exe/fetch.php?media=intern:abschlussarbeiten:2012:
  ma2012_hausmann_petra.pdf, 2012.

---

## Author Comment (AC2) · 8 Aug 2016

Dear Reviewer,

thank you for your support and for your suggestions for improving our manuscript.

In general, all reviewers suggest to strengthen the literature review, especially to improve the discussion of earlier publications on imaging ice cloud remote sensing (Schaefer et al. 2013) and the introduction of the visible spectral slope solution for the transmissivity ambiguity (Brueckner el al. 2014 and Le Blanc et al 2015). This is an obvious weakness of our manuscript. The reason for this negligence on our side is partly due to the fact that our manuscript has had a long history already. In our group the spectral slope approach originally goes back to a Master's thesis of co-author Petra Hausmann from 2012. We obviously noticed that "our approach" was published meanwhile in proper journals by others. Even though this is no excuse for gaps in our literature review, it might explain why we do not want to state any direct "use" or "application" of ideas introduced by the aforementioned authors. In our revision we do both, we try to strengthen our literature discussion, and at the same time we would like to include the Hausmann Master's thesis from 2012 as a reference. Although it is no peer-reviewed publication it is an official university thesis in English language available online.

Point by point reply to all major comments (all minor were considered as suggested apart from the ones mentioned below):

**Major comments**

1. **Literature review:** One main issue of the manuscript is the insufficient literature review and comparison to recently published studies. This concerns in particular the handling of the ambiguity between transmitted solar spectral radiance and cloud optical thickness as well as to comparisons of the results of the sensitivity study to literature values.

    a. **Ambiguity:** In the current manuscript, a third dimension is applied to the classical two-wavelength cloud retrieval by Nakajima and King (1990) to avoid the ambiguity between transmitted solar spectral radiance and cloud optical thickness. This third dimension is given by a slope-fit/ratio in the visible wavelength range between 485 and 560 nm. Recently, Brückner et al. (2014) published a similar method using ratios in the visible wavelength range for the third dimension. The method presented by Brückner et al. (2014) definitely should be considered and discussed in the current manuscript.

    => The section in the introduction now reads:

    ```
    Recently Brückner et al. (2014) as well as LeBlanc et al. (2015)
    presented similar solutions for unambiguous retrievals of optical
    thickness and effective radius for pointing system without
    providing imagery. Both suggest the use of spectral slopes in the
    visible to separate between the two optical thickness regimes.
    We will present a combination of both, a solution for the
    transmittance ambiguity using a similar spectral slope (following
    ideas of Hausmann, 2012) and results for imaging measurements which
    provide context information on the distribution of optical
    thickness and effective radius over a large area.
    ```

b. **Sensitivity study:** The present manuscript provides a detailed and impressive sensitive study on possible retrieval uncertainties. However, the results should be compared to the results of the sensitivity study given by Schäfer et al. (2013). For a cirrus retrieval adapted to the measurements with a ground-based imaging spectrometer in the visible wavelength range, Schäfer et al. (2013) investigated the retrieval uncertainties of cloud optical thickness retrievals, e. g. including surface albedo and cirrus crystal shape.

We also mention Brückner and LeBlanc at the end of section 3 "Retrieval..." where we presented our version of the idea and in the section 5 "Summary and Discussion".
=> We included a comparison to the Schäfer et al results into the discussion section:

```
Schäfer et al. (2013) also assessed the sensitivity of their
ground-based cirrus optical thickness retrieval to variation of
certain parameters. The values can not be directly compared to our
results, as they only refer to a small number of specific
situations regarding observation geometry and cirrus situation and
not a large range of combinations as in our sensitivity test. For
variation of crystal habit and for small optical thickness up to 1
they showed large relative differences up to 80% with average
absolute differences at 0.1. Though such cases are contained in the
sensitivity test shown here, average impact over many different
situations is smaller. Schäfer et al. (2013) also present large
uncertainties for an albedo variation. This is caused by their
choice of a test albedo which is extremely different from the
measurement situation, while here it was assumed that the general
albedo situation can be characterized well and remaining
uncertainty has only small impact.
```

2. **2$^{nd}$ test case:** From my point of view, the discussion of the second test case from 2 October 2012 should be removed from the manuscript. The first case is already sufficient to demonstrate the ability of the introduced cirrus retrieval to give proper results. The manuscript will not benefit from the second case. Of course, it would be nice to have two satellite products to compare, but due to the contamination by low clouds, a comparison will not be significant. Furthermore, the data seem to be overexposed at multiple parts of the image, which may be the reason that no cloud retrieval could be adapted at those parts.

=> We think the second test case should stay in the manuscript.

(1) There is no overexposure in the data. For figure part (a) the color scale was cut at 0.5. We corrected that.

(2) We do not only want to show a single perfect example, but also show an example where the quality is not so good for good reasons ("quality" was renamed "significance" following a comment from another reviewer).

(3) This second example is also interesting because it demonstrates the possible advantages of a ground-based method. Looking upward, clouds below the mountain top do not directly affect the retrieval, except that they increase the albedo (in contrast to the satellite retrievals which obviously are affected). Very likely our results are the best possibility to provide a "ground truth" for cirrus satellite retrievals in such situations. The possible implications of albedo changes by the underlying cloud patches around the sensor

position are also discussed in an additional "spectral albedo" test case in the sensitivity tests and mentioned for this example. We discuss that in the end of this section:

```
An interesting aspect of this complex example is the demonstration
of the potential of a ground-based method to provide accurate cloud
properties compared to satellite methods, especially for thin
cirrus. The same quantities are retrieved by both methods,
utilising similar wavelength bands, but the ground-based method
benefits from its much higher spatial resolution which allows to
separate different parts (or layers) of the observed cloudiness. In
the ground-based data there might still be an impact of increased
albedo (low level cumulus below the instrument). The low levels of
significance of our results at larger sensor zenith angles might be
a sign of it (see Fig. 12d). Nonetheless the ground-based method is
less affected by this problem and generally most likely much better
at retrieving thin ice cloud properties than the satellite methods.
```

**Minor and technical comments**

1. **Acronyms**: Acronyms are often used several times before they were introduced the first time. Examples are LMU, specMACS, ACRIDICON, MODIS, SEVIRI, CloudSat, CALIPSO. I don't know if I got them all. Please check all acronyms throughout the manuscript and introduce their full names whenever they are used for the first time.

2. **Indices and units:** Indices and units are sometimes written in italic letters and sometimes in non-italic letters. Throughout the manuscript this happens also for one and the same index or unit. For reasons of consistency you should write all indices and units in non-italic letters.

=>Acronyms: I tried to introduce all acronyms twice, in the abstract and in the main text as required by AMT guidelines. Unfortunately that leads to unreadable sentences in the abstract. I will leave a comment on that to the Copernicus type setting and ask them to find an acceptable solution.

=> Indices and units: Following the AMT guidelines equations and mathematical symbols should be in italic letters. I think this is also true for equation parts in the text. Units should not and I checked these.

11. **Fig. 8, 9, 12:** Please increase font size

=> I increased font size of Fig 8. For Figure 9 and 12 I would prefer a larger image size which also will depend on the later layout.

**Reference:**

- Hausmann, P.: Ground-based remote sensing of optically thin ice clouds, 89 pages, Master's thesis, Ludwig-Maximilians-Universität, Munich, http://www.meteo.physik.uni-muenchen.de/DokuWiki/lib/exe/fetch.php?media=intern:abschlussarbeiten:2012:ma2012_hausmann_petra.pdf, 2012.

---

## Author Comment (AC3) · 8 Aug 2016

Dear Reviewer,

thank you for your support and for your suggestions for improving our manuscript.

In general, all reviewers suggest to strengthen the literature review, especially to improve the discussion of earlier publications on imaging ice cloud remote sensing (Schaefer et al. 2013) and the introduction of the visible spectral slope solution for the transmissivity ambiguity (Brueckner el al. 2014 and Le Blanc et al 2015). This is an obvious weakness of our manuscript. The reason for this negligence on our side is partly due to the fact that our manuscript has had a long history already. In our group the spectral slope approach originally goes back to a Master's thesis of co-author Petra Hausmann from 2012. We obviously noticed that "our approach" was published meanwhile in proper journals by others. Even though this is no excuse for gaps in our literature review, it might explain why we do not want to state any direct "use" or "application" of ideas introduced by the aforementioned authors. In our revision we do both, we try to strengthen our literature discussion, and at the same time we would like to include the Hausmann Master's thesis from 2012 as a reference. Although it is no peer-reviewed publication it is an official university thesis in English language available online.

Point by point reply to all major comments (all minor were considered as suggested):

Sequential major comments:

P2, L27: King et al. (2004) is the wrong reference for the adaptation of the Nakajima/King algorithm to ice clouds. At this point, the MODIS retrievals were already fully operational. One of the Platnick references is probably more appropriate; the cited paper is for the retrievals over bright surfaces, an entirely different problem. A nice review of ice cloud retrievals (with emphasis on the problematic phase function) is provided by *Wang et al.* (2014).

> => I once more tried to pin down the best "original" publication for our statement that Nakajima+King „can be adapted to ice". Even in the MODIS ATBD this King et al publication is listed at this point. We now mention Baum et al. 2000 on MODIS airborne simulator and ice properties which is slightly older and maybe more general. We also added the aspect of TIR cirrus remote sensing mentioning Wang et al. 2011 to the introduction. Thanks for pointing out.

P3, L16: Clarify (here or later) that *LeBlanc et al.* (2015) used the spectral slope of normalised mid-visible transmittance (or radiance), and that this is what the algorithm in this manuscript relies on as well (rather than using the *combination* of the spectral slope and imaging capabilities, as the subsequent sentence implies). See also the comments on pages 1 and 2 of this review. Also change this appropriately at later occurrences (for example in the summary); it should be stated that the idea for resolving the ambiguity problem was first introduced by the 2014 and 2015 papers.

> => The section in the introduction now reads:

> ```
> Recently Brückner et al. (2014) as well as LeBlanc et al. (2015)
> presented similar solutions for unambiguous retrievals of optical
> thickness and effective radius for pointing system without
> providing imagery. Both suggest the use of spectral slopes in the
> visible to separate between the two optical thickness regimes.
> We will present a combination of both, a solution for the
> transmittance ambiguity using a similar spectral slope (following
> ```

ideas of Hausmann, 2012) and results for imaging measurements which
provide context information on the distribution of optical
thickness and effective radius over a large area.

P4, L24: 16 streams seem too few to properly resolve the features of the scattering
phase function unless the cited intensity correction works properly for small and
medium optical thickness values (single or low-order scattering) – this is of course
especially true for scattering angles near the halo angles or other pronounced
features. It would be highly recommended to show a plot of simulated downwelling
radiance as a function of viewing angle such that the ability of the RT code to
simulate the halo in the right place becomes credible. Too few streams in the solver
may misplace the halo despite the application of the intensity correction. Perhaps
such a plot could be added in an appendix?

=> We have been aware of this pitfall. We think that we are able to resolve the phase
function correctly.  This image shows a simulation of the 22° halo for pristine solid column
ice particles simulated for an
almucantar scan of a sun-
photometer using an exact method
(MYSTIC Monte Carlo) and 16-
streams libRadtran DISORT.

[Figure]

Nonetheless we in general do not
suggest the retrieval for scattering
angles around the 22 and 46° halos,
because of the variations of the
intensity of such features due to
crystal shape assumptions (and
crystal roughness). With an imager
we are almost always able to avoid
this region and our example
applications do not include such
geometries.

I would prefer not to show a plot similar to the above in the manuscript, as it is a rather
technical aspect and we avoid these scattering angles anyway.

We extended the final discussion of these points:

Of course the most important step forward would consist in a
reduction of the crystal type uncertainty. The halo regions around
22° and 46° scattering angle were avoided here for our spectral
approach. Uncertainties can be expected to be higher in these
regions with strong angular gradients of transmittance under single
scattering conditions, if no additional information on crystal
habits is available. However the imaging capabilities of the
specMACS sensor (especially if combined with a scanning platform,
see Ewald et al., 2015) do not only allow to successfully avoid
these regions for the spectral evaluation, but would allow for the
utilization of the spatial distribution of transmittance in these
regions to provide the missing information. Use of this spatial
distribution could provide important constraints regarding the
present average phase function as Schaefer et al  (2013)
demonstrated. Especially the presence of optical scattering
phenomena like type and intensity of halo displays could be used to

identify specific particle shapes and orientation and information
on the mixture with less perfect rough ice particles. A combination
of the presented method with additional information of this kind
will be the next step in our effort to provide better ice cloud
property observations.

Section 2.2.2, optical properties: Using all six habits from "HEY" may not be
appropriate since some of the habits are highly unlikely to occur in the study region.
The authors do mention that the errors due to crystal shape represent an upper
bound for this reason, but even that may not be correct if a certain habit does not
occur at all in the kind of cloud that is observed. This point, combined with the
previous one (are the features of the phase functions are resolved by the intensity-
corrected 16-stream radiances?) cast some doubt on the concluded magnitude of
the bias due to unknown habit.

=> As the true ice particle habit is never known, there is no other thing we can do but
discuss maximum errors? Also we are far from identifying typical habit mixtures or typical
crystal roughness for specific regions of the world. Even the long established mixtures like
the Baum mixture are about to be abandoned (e.g., MODIS group) in favor of simple single
habits with added roughness. So I do not see an easy way to improve the situation, before
methods similar to the Schaefer et al one are developed for robust operational use. We
extended the discussion along this line (see above).

Section 2.2.3, surface albedo: MODIS surface albedo products are known to have
problems in regions with pronounced topography. Given that, why was the
sensitivity to surface albedo and its uncertainty not tested in similar ways as done
for the other parameters (see table 1)? This seems even more important than
testing the sensitivity to the presence of aerosols.

=> Correct comment. Originally we did not test albedo, because we use a MODIS albedo
product for the time of measurement. We now included a test of the sensitivity towards
inaccuracies there with a second spectral albedo set. We added text in the sensitivity test
section:

As an actual spectral albedo is averaged in a MODIS product over 16
days including the measurement period, the correct albedo influence
should be represented in our forward simulations. Still
uncertainties arise from the products uncertainty, the derivation
process of the albedo data for the Zugspitze area and vegetation
changes during the period. As test case, a second albedo data set
for our main measurement site in the center of Munich city area is
used. It is shown in lighter colors in Fig. 2. In the urban area,
the vegetation peak between 750 nm and 1400 nm is much more
pronounced (Munich is a green city), while the changes at shorter
and longer wavelength are smaller. Nonetheless, the Munich albedo
data set is also brighter in the regions used for the retrieval
around 550 nm and 1600 nm by 15 and 20%. This increase is
comparable to the difference between summer and winter for
vegetated surfaces and is, at the same time, much larger than the
estimate of the albedo product uncertainty (see Moody et al.,
2005). Errors and uncertainties caused by such a variation of
albedo are, nonetheless, small for reff (bias at 0.3 an RMSE at 4
um) and tau (no bias, RMSE at 0.5).

P6, Eqn (2) This parameter is equivalent to $\eta_{11}$ from *LeBlanc et al.* (2015), except that they used radiance (not transmittance) to derive it, which is conceptually the same – see also previous comments.

P6,l24: "Consequently, it is *now* possible to…" See comments above (origin/genesis).

=> See reply above. The word „now" was removed and a hint to the Brueckner and LeBlanc publications is left in the end of our description.

P7,l10: Why does the LUT only include three different relative azimuth angles? Can this be justified by the geometric arrangement and a special sun-viewing geometry? The relevant angle for near-single scattering remote sensing (i.e., up to about 1 optical thickness) should be the scattering angle, not the viewing zenith + azimuth angle, correct? This may only be a minor point – however, typical LUTs usually feature more than the azimuth angles used in this study, so the question is why this study can get by with so few. A plot that shows the dependence of the radiance on viewing azimuth angle for a small optical thickness would help giving some credence to the approach. Combined with an earlier suggestion: The approach can only be regarded as sound if the halo (or another feature) can be simulated in the right place (in the 2D image) and resolved both in viewing zenith and azimuth angle.

=> Before measurements the alignment of the instrument was carried out in a very simple fashion. The sensor was leveled pointing into zenith and a simple sun shadow system was used to fix the position of the spatial line perpendicular or in parallel to the solar azimuth. Thus only these geometries could appear in our measurements. New text in that section:

```
Since the sensor was aligned with respect to the sun before each
measurement, a very limited set of viewing zenith and azimuth
angles has to be provided in the lookup table. ….
… Three relative azimuth angles are taken into account, because the
sensor's spatial line was either aligned parallel or perpendicular
to the principal plane: phi_rel = 0° and phi_rel = 180° (line
parallel) or phi_rel = 90° (perpendicular).
```

P8,Eqn 3: It is unclear why the three terms (T550, T1600, SVIS) are each given equal weight in determining dT. After all, the measurement uncertainties propagate into each of these in different ways, and the individual terms should be weighed by their (different) uncertainty, rather than applying a global weight for the different LUT points as proposed in Eqn 5 later on.

=> The overall range of values is similar for all three parameters. A global rescaling would not change much. We have originally tried a re-scaling of the lookup table using local gradients of measured parameters with changing reff and tau and the use of a dT normalised this way. According to our sensitivity test suite, the improvement of result accuracy was similar to the one that we could reach by increasing the density of points in the lookup table. For the sake of simplicity we decided to use slightly more computational effort and abandon the more sophisticated version.

P8/P9: Related to the comment above: How is the uncertainty in COD and REFF derived from the uncertainty in the measurements (the radiances)? dT cannot be regarded as a metric for the uncertainty because it is a merely technical quantity which would change for a different LUT gridding. Also, trading standard deviation for uncertainty as done on p12,l20-21 does not help. How do we know for certain that the error does not exceed the standard deviation? Particularly for small crystal size (on the order of 10-20 micron) and the optical thickness range shown, the effective radius may be highly uncertain because of the issues described by *McBride et al.* (2011). An error propagation analysis from the radiances to the retrieval parameters should be performed, not in terms of the LUT gridding, but in terms of the underlying physics. This is probably the most work-intensive revision that is suggested in this review.

=> Right, such a complete error propagation analysis would be very valuable. Unfortunately it goes beyond the effort we can provide for this manuscript at the moment. We strengthened the discussion of your points and replace the term "quality" by "significance" in this context.

New text were the parameter is defined:

```
Significance 1-deltaT /  deltaTmax=0 is related to the maximum
search radius, while larger values are related to better matches
and a perfect match would be significance 1.The term significance
is chosen because the distance to the tabulated values, strictly
speaking, is not a measure of retrieval accuracy or quality, but
rather a technical quantity. Given the fact that we expect matching
measured and tabulated values, if we have considered the influence
factors correctly, deltaT is an important parameter which indicates
the applicability of the chosen lookup table to the real
measurement situation and consequently the reliability of the
results. Still this parameter could be small for the wrong reasons,
e.g., if impact of wrong albedo and wrong ice particle habit
compensate each other.
```

New text in the discussion of October 3 results:

```
Retrievals have been possible at retrieval significance values
close to 0.75 and above throughout the largest part of the scene.
Following our definition of this value this means that for all
pixel measurements of spectral transmissivity tabulated values are
consistently within deltaT=0.025  and 0.01 of the tabulated lookup
values' surface (compare Fig. 5).The real situation seems to be
realistically represented by the lookup table.
```

P10,L9: See earlier comment concerning the azimuth vs. scattering angle. Azimuth does not seem to be the appropriate parameter to consider for singe-scattering conditions.

=> As mentioned in the presentation of the method there is only one set of lookup tables for a given measurement situation which has to b used, because the relative azimuth is fixed. Then the set of lookup tables with most comparable zenith angles is selected and interpolated using the scattering angle only. So you are probably right, using scattering angle as main parameter for the lookup tables might be more efficient. But first, we are sure that results would be the same (due to the last interpolation step) and second, we do not want to limited ourselves to a single scattering regime here.

P12,l20-21: See earlier comment: The standard deviation cannot replace a physics-based error analysis. Also: The impact of surface albedo uncertainty should be part of the retrieval error, unless it is deemed to be negligible.

> => See our earlier replies.

P10,l22-23: Would it help to limit to such habits that can realistically occur for the type of cloud under consideration? What about the distinction between habit and surface roughness (which seems to have a larger role than the habit when it comes to smooth versus "featured" phase functions)? Is it possible to say something about the roughness using the imaging capabilities of the instrument (in the vein of the Schäfer paper)?

> => As said before, I think there is no way to limit the habit possibilities beforehand with a spectral method only (as presented in this paper). But you are right, a further development of ideas like the Schaefer et al methods using the spatial distribution of transmittance over a wider range of scattering angles would make this possible and is definitely among our goals going beyond this paper.

P14,l16 & l18-20: Please consider earlier comments regarding the origin of the "new third parameter" for the removal of the ambiguity.

> => We mention Brueckner and Leblanc once more.

P16,l1-5: I commend the authors on tackling the arduous task to reduce the retrieval uncertainty due to crystal shape using spectral imaging!

> => We are working on it! For a further publication.

**References:**

- Baum, B. A., Kratz, D. P., Yang, P., Ou, S. C., Hu, Y., Soulen, P. F., and Tsay, S.-C.: Remote sensing of cloud properties using MODIS airborne simulator imagery during SUCCESS: 1. Data and models, Journal of Geophysical Research: Atmospheres, 105, 11 767–11 780, doi:10.1029/1999JD901089, http://dx.doi.org/10.1029/1999JD901089, 2000.

- Hausmann, P.: Ground-based remote sensing of optically thin ice clouds, 89 pages, Master's thesis, Ludwig-Maximilians-Universität, Munich, http://www.meteo.physik.uni-muenchen.de/DokuWiki/lib/exe/fetch.php?media=intern:abschlussarbeiten:2012:ma2012_hausmann_petra.pdf, 2012.

- Moody, E. G., King, M. D., Platnick, S., Schaaf, C. B., and Gao, F.: Spatially complete global spectral surface albedos: value-added datasets derived from Terra MODIS land products, IEEE Transactions on Geoscience and Remote Sensing, 43, 144–158, doi:10.1109/TGRS.2004.838359, 2005.